

# The ecomorphology of the shell of extant turtles and its applications for fossil turtles

Laura Dziomber[1,2], Walter G. Joyce[1] and Christian Foth[1]

[1] Department of Geosciences, University of Fribourg, Fribourg, Switzerland
[2] Institute of Plant Sciences & Oeschger Centre for Climate Change Research, University of Bern, Bern, Switzerland

## ABSTRACT

Turtles are a successful clade of reptiles that originated in the Late Triassic. The group adapted during its evolution to different types of environments, ranging from dry land to ponds, rivers, and the open ocean, and survived all Mesozoic and Cenozoic extinction events. The body of turtles is characterized by a shell, which has been hypothesized to have several biological roles, like protection, thermal and pH regulation, but also to be adapted in its shape to the ecology of the animal. However, only few studies have investigated the relationships between shell shape and ecology in a global context or clarified if shape can be used to diagnose habitat preferences in fossil representatives. Here, we assembled a three-dimensional dataset of 69 extant turtles and three fossils, in particular, the Late Triassic *Proganochelys quenstedtii* and *Proterochersis robusta* and the Late Jurassic *Plesiochelys bigleri* to test explicitly for a relationship between shell shape and ecology. 3D models were obtained using surface scanning and photogrammetry. The general shape of the shells was captured using geometric morphometrics. The habitat ecology of extant turtles was classified using the webbing of their forelimbs as a proxy. Principal component analysis (PCA) highlights much overlap between habitat groups. Discriminant analyses suggests significant differences between extant terrestrial turtles, extant fully aquatic (i.e., marine and riverine) turtles, and an unspecialized assemblage that includes extant turtles from all habitats, mostly freshwater aquatic forms. The paleoecology of the three fossil species cannot be determined with confidence, as all three fall within the unspecialized category, even if *Plesiochelys bigleri* plots closer to fully aquatic turtles, while the two Triassic species group closer to extant terrestrial forms. Although the shape of the shell of turtles indeed contains an ecological signal, it is overall too weak to uncover using shell shape in paleoecological studies, at least with the methods we selected.

Corresponding author
Laura Dziomber,
laura.dziomber@ips.unibe.ch

## INTRODUCTION

Turtles represent a remarkable group of tetrapods due to the presence of an ossified shell. The clade Testudinata (*sensu Joyce, Parham & Gauthier, 2004*) is defined by the presence of this trait and is represented by more than 350 extant species (*Turtle Taxonomy Working*

*Group, 2017*) and a rich fossil record that reaches back to the Late Triassic (*Młynarski, 1976*). A number of other groups of tetrapods convergently acquired an armored body plan as well, in particular armadillos (*Chen et al., 2011*), ankylosaurs (*Hayashi et al., 2010*), aetosaurs (*Desojo et al., 2013*), and placodonts (*Westphal, 1976*), but none have proven to be particularly successful, at least in regard to phylogenetic longevity, biogeographic distribution, diversity, or disparity.

## The turtle shell

The shell is a common characteristic of all turtles but subject to substantial morphological variation from one species to the other (*Pritchard, 2008*). It is universally composed of the dorsal carapace and the ventral plastron. From an anatomical perspective, the shell is a composite of the dermis with underlying, preexisting structures, in particular the dorsal ribs, dorsal vertebrae, gastralia, the clavicle, interclavicle, and cleithra (*Lyson et al., 2013a*; *Lyson et al., 2013b*). The resulting bones of the carapace of a typical turtle are called the neurals, costals, nuchal, peripherals, and pygals (Fig. 1D), those of the plastron the entoplastron and the epi-, hyo-, meso-, hypo-, and xiphiplastra (*Zangerl, 1969*, Fig. 1E). The bony shell is protected towards the outside by a layer of keratinous, epidermal scutes, but these are secondarily reduced in trionychids (softshell turtles), carettochelyids (pig-nosed turtles), and dermochelyids (leatherback turtles). The scutes of the carapace of a typical turtle are termed cervicals, vertebrals, pleurals, and marginals (Fig. 1D), and those of the plastron gulars, extragulars, humerals, pectorals, abdominals, femorals, and anals (*Zangerl, 1969*; *Hutchison & Bramble, 1981*, Fig. 1E). The number and the contacts of the bony and epidermal elements vary immensely across turtles and can both be used to diagnose species and to reconstruct phylogenetic relationships. It is therefore not surprising that a large body of literature is dedicated to documenting this type of variation to the turtle shell.

The turtle shell is thought to provide several evolutionary advantages, including protection, pH control, or thermal regulation (*Jackson, 2000*; *Pritchard, 2008*; *Magwene & Socha, 2013*). The presence of this full body armor, however, is thought to constrain other bodily functions, in particular feeding, locomotion, reproduction, and respiration. A number of shell shapes have developed as a compromise. For instance, teardrop-shaped shells (e.g., the chelonioid *Chelonia mydas*) are more typical for turtles with aquatic habits, especially those that live in open marine environments (*Wyneken, 1996*), while highly domed shells (e.g., the testudinid *Stigmochelys pardalis*) are prevalent among turtles with terrestrial habitats (*Domokos & Várkonyi, 2008*). A large diversity of additional morphologies can be observed, including the oval and tectiform shells of many riverine turtles (e.g., emydid *Graptemys geographica*) or the rounded and greatly flattened shells of many trionychids (e.g., *Apalone spinifera*). Given that correlations appear to exist between shell shape and ecology, paleontologists have historically been tempted to reconstruct the paleoecology of fossil turtles by reference to their shell shape, but studies have been lacking that explicitly tested this relationship.

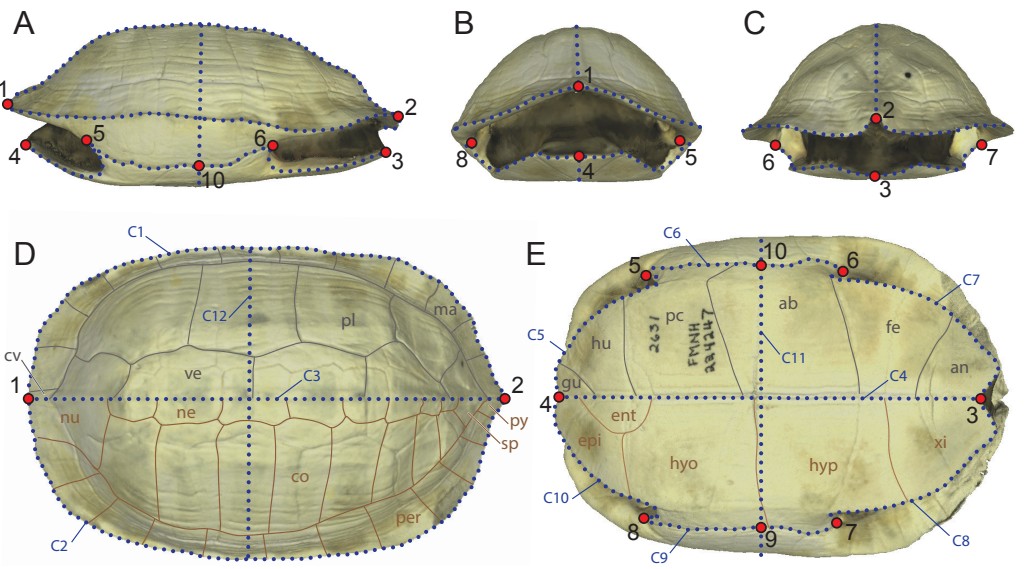

**Figure 1** Landmarks configuration used in the study composed of 10 fixed landmarks and 12 semilandmark-curves imposed onto a 3D model of *Melanochelys trijuga* (FMNH 224247). (A) Left lateral view. (B) Anterior view. (C) Posterior view. (D) Dorsal view (grey: epidermal scutes; brown: dermal bones). (E) Ventral view. (grey: epidermal scutes; brown: dermal bones). Abbreviations: ab, abdominal scute; an, anal scute; cv, cervical scute; co, costal; ent, entoplastron; epi, epiplastron; fe, femoral scute; gu, gular scute; hu, humeral scute; hyp, hypoplastron; hyo, hyoplastron; ma, marginal scute; ne, neural; nu, nuchal; pe, pectoral scute; pl, pleural scute; per, peripheral; py, pygal; sp, suprapygal; xi, xiphiplastron.

## Morphometrics in turtles

A broad selection of studies have recently focused on finding correlations between the ecology of extant turtles and their cranial or post-cranial morphology, including morphometrics (e.g., *Joyce & Gauthier, 2004*; *Domokos & Várkonyi, 2008*; *Benson et al., 2011*; *Lichtig & Lucas, 2017*), histology (e.g., *Scheyer & Sander, 2007*), geometric morphometrics (e.g., *Claude et al., 2003*; *Claude et al., 2004*; *Depecker et al., 2006*; *Rivera, 2008*; *Rivera & Claude, 2008*; *Stayton, 2011*; *Foth, Rabi & Joyce, 2017*; *Foth et al., 2019*), and Finite Element Analysis (e.g., *Stayton, 2009*; *Polly et al., 2016*). A number of these studies were performed with the explicit goal of finding correlations among extant turtles to reconstruct the paleoecology of the oldest known fossil turtles, a topic with considerable interest regarding the origin and early evolution of the group.

Two taxa have been at the center of these studies: *Proganochelys quenstedtii Baur, 1887* and *Proterochersis robusta Fraas, 1913* from the Late Triassic of Germany. *Proganochelys quenstedtii* was originally argued to have had been a fresh-water aquatic bottom walker based on its low shell and details in femoral anatomy (*Gaffney, 1990*), while *Proterochersis robusta* was tacitly assumed to be terrestrial based on its highly domed shell (e.g., *Fraas, 1913*; *De Lapparent de Broin, 2001*). *Joyce & Gauthier (2004)* used morphometric measurements from forelimb bones, in particular the relative length of the humerus, ulna, and hand, as a proxy for the habitat preferences of extant and fossil turtles. For this study, extant turtles were classified into six different ecological categories ranging from completely terrestrial to
completely aquatic. The data shows a strong correlation between the relative length of the hand and the ecology of extant turtles, with terrestrial turtles having shorter hands than aquatic ones, and predicts *Proganochelys quenstedtii* to have been terrestrial. *Proterochersis robusta* was not included in this study, as its forelimbs are not preserved. This conclusion was broadly corroborated by *Scheyer & Sander (2007)*, who noted through a study of bone histology that the bone microstructure of *Proganochelys quenstedtii* and *Proterochersis robusta* more closely resembles that of extant terrestrial turtles than that of extant aquatic turtles. *Benson et al. (2011)* concluded based on shell cross-section morphometrics of the shell of extant turtles, as quantified from photographs, that *Proterochersis robusta* was likely semi-aquatic, although it is important to note that the habitat categories of *Benson et al. (2011)* do not overlap with those of *Joyce & Gauthier (2004)*. The recent study of *Lichtig & Lucas (2017)*, finally, inferred a freshwater aquatic ecology for *Proganochelys quenstedtii* and a terrestrial ecology for *Proterochersis robusta* using ratios from the shell, in particular maximum carapace width to maximum plastron and carapace length to maximum carapace height. It therefore appears that different lines of evidence yield conflicting results.

### Aims of the study

Previous studies that assessed the ecology of fossil turtles using the turtle shell as a source of information only utilized selected aspects of the shell. The initial aim of this study is to first test for correlations between ecology and the entire shell shape of extant turtles, using three-dimensional geometric morphometrics in combination with multivariate analyses. The correlations observed among extant turtles are then applied to the Late Triassic turtles *Proganochelys quenstedtii* and *Proterochersis robusta* and the Late Jurassic turtle *Plesiochelys bigleri*.

## MATERIAL AND METHODS

### Taxonomic sampling

The sample of extant turtles includes species representing all turtle clades and habitat preferences. Sampling was strictly limited to specimens collected as adults from the wild, as the shell of many turtles grows into an unnatural shape when kept in captivity, such as the pyramidal scutes seen in captive-raised tortoises (*Wiesner & Iben, 2003*; *Gerlach, 2004*). To avoid biases caused by sampling different ontogenetic stages, sampling was furthermore restricted to skeletally mature individuals. The sole exception to this rule is the giant leatherback turtle *Dermochelys coriacea*, the only representative of its clade, for which a juvenile specimen was chosen (carapace length ca. 13 cm), since no intact adult specimens were available for this study. Finally, sampling was limited to specimens with complete shells, including naturally articulated bridges, that lack scute abnormalities, shell deformations (e.g., kyphosis), or pronounced asymmetry. Sex was disregarded as a selection criterion, as most specimens housed in collections, especially skeletal specimens, are not sexed and as the sex of turtles is only known to influence the overall shape of the shell in a subtle manner (*Pritchard, 2008*). To substantially increase sample size, specimens were included with varying preservation methods, including dry skeletal specimens, mummified specimens, and specimens conserved in ethanol. The inclusion of ethanol

preserved individuals particularly allowed sampling trionychids and the leatherback turtle *Dermochelys coriacea.*

To optimize phylogenetic coverage, we attempted to sample at least one species of each currently recognized genus of extant turtle (*TTWG, 2017*). Several species were sampled, however, for genera that exhibit ecological plasticity, in particular *Cuora, Terrapene,* and *Rhinoclemmys,* genera that contain both aquatic and terrestrial species. The final primary dataset consists of 69 species of extant turtles (see Table 1) that represent all major turtle clades. Generic sampling exceeds 50% for all clades but Podocnemididae (detailed in Table S1).

In addition to recent turtles, the sample furthermore includes three species of fossil turtles: the thalassochelydian *Plesiochelys bigleri Püntener, Anquetin & Billon-Bruyat, 2017* from the Late Jurassic of Switzerland, *Proganochelys quenstedtii* from the Late Triassic of Germany and *Proterochersis robusta* from the Late Triassic of Germany. For the fossil turtles, the best-preserved specimens were chosen to represent each species (see Table 1), except in the case of *Proganochelys quenstedtii,* for which a cast of SMNS 16980 was scanned (*Gaffney, 1990*).

## Acquisition of 3D models

We generated 3D models of turtle shells using two main techniques. The 3D scanner *Artec Space Spider,* which produces 3D models utilizing structured light, was used for most specimens with a length smaller than 60 cm. The reconstruction of the models was done using *Artec Studio Professional 10.* Larger specimens were sampled using close-range photogrammetry. The models obtained were computed using the software *Agisoft Photoscan Professional* based on photographs taken with an *Olympus E-M10* camera. All 3D models were generated by us, expect the one of *Plesiochelys bigleri,* which was made available by *Raselli & Anquetin (2019a)* and *Raselli & Anquetin (2019b).* All 3D models reconstructed by us for this project are available on MorphoSource (*Dziomber, Joyce & Foth, 2020*; see Table S2 for the DOI of these specimens).

## Morphometric measurements

Some of the previous geometric morphometric studies of the turtle shell attempted to capture its morphology by utilizing as many type-I landmarks as possible, in particular those created by the contacts of the bones and the overlying epidermal scutes (e.g., *Claude et al., 2003*; *Angielczyk & Sheets, 2007*). As various groups of turtles lack all or some dermal bones or epidermal scutes (e.g., carettochelyid, dermochelyids, trionychids), use of type-I landmarks defined by these structures precludes utilizing the full spectrum of morphotypes developed by turtles over the course of their history. In addition, as the shape and placement of the bones and epidermal scutes on the shell of a turtle are strongly influenced by phylogenetic history, use of type-I landmarks defined by these structures is optimal for capturing the phylogenetic information held by the subparts of the shell, not the shape of the shell in itself. We therefore here implement an alternative approach that uses a set of ten homologous landmarks and 255 semilandmarks distributed on twelve curves (Fig. 1). The landmarks represent geometric points, in particular the anterior-most
**Table 1 Composition of the extant turtles included in the dataset of this study.** Every specimen is associated with a clade, a species name, catalog number, type of preservation (Pres.), ecological category (Cat.) based on webbing ranging (0 to 4) (see Methods) and acquisition method (Acq.).

| Clade | Species | Catalog Number | Pres | Cat | Acq |
|---|---|---|---|---|---|
| Carettochelyidae | *Carettochelys insculpta* | FMNH 15480 | DRS | 4 | SC |
| Chelidae | *Platemys platycephala* | FMNH 267453 | ETH | 1 | SC |
| Chelidae | *Chelus fimbriata* | FMNH 250681 | DRS | 2 | SC |
| Chelidae | *Mesoclemmys dahli* | FMNH 82302 | DRS | 2 | SC |
| Chelidae | *Phrynops tuberosus* | FMNH 73434 | DRS | 2 | SC |
| Chelidae | *Elseya novaeguineae* | FMNH 14054 | DRS | 2 | SC |
| Chelidae | *Emydura macquarii* | FMNH 71793 | ETH | 2 | SC |
| Chelidae | *Hydromedusa tectifera* | FMNH 217272 | ETH | 3 | SC |
| Chelidae | *Chelodina oblonga* | FMNH 77997 | ETH | 3 | SC |
| Cheloniidae | *Chelonia mydas* | NMB 152 | ETH | 4 | PH |
| Cheloniidae | *Caretta caretta* | MHNF 11858_1993 | ETH | 4 | PH |
| Cheloniidae | *Eretmochelys imbricata* | NMB 5763 | ETH | 4 | PH |
| Chelydridae | *Macrochelys temminckii* | NMB 14 | MUM | 2 | PH |
| Chelydridae | *Chelydra serpentina* | FMNH 14710 | DRS | 2 | SC |
| Dermatemydidae | *Dermatemys mawii* | FMNH 4163 | DRS | 2 | SC |
| Dermochelyidae | *Dermochelys coriacea* | FMNH 61630 | ETH | 4 | SC |
| Emydidae | *Trachemys scripta* | FMNH 268818 | DRS | 2 | SC |
| Emydidae | *Terrapene carolina* | FMNH 211600 | DRS | 0 | SC |
| Emydidae | *Clemmys guttata* | FMNH 83369 | DRS | 1 | SC |
| Emydidae | *Emys orbicularis* | FMNH 15654 | MUM | 1 | SC |
| Emydidae | *Glyptemys insculpta* | FMNH 283801 | DRS | 1 | SC |
| Emydidae | *Emys blandingii* | FMNH 83439 | DRS | 1 | SC |
| Emydidae | *Deirochelys reticularia* | FMNH 83401 | DRS | 2 | SC |
| Emydidae | *Graptemys geographica* | FMNH 83367 | DRS | 2 | SC |
| Emydidae | *Malaclemys terrapin* | FMNH 83411 | DRS | 2 | SC |
| Emydidae | *Chrysemys picta* | FMNH 242270 | DRS | 2 | SC |
| Emydidae | *Actinemys marmorata* | FMNH 211580 | DRS | 2 | SC |
| Geoemydidae | *Geoemyda spengleri* | FMNH 260381 | DRS | 0 | SC |
| Geoemydidae | *Vijayachelys silvatica* | FMNH 224155 | ETH | 0 | SC |
| Geoemydidae | *Rhinoclemmys annulata* | FMNH 63923 | DRS | 1 | SC |
| Geoemydidae | *Cuora amboinensis* | FMNH 224028 | DRS | 2 | SC |
| Geoemydidae | *Cyclemys dentata* | FMNH 224085 | DRS | 2 | SC |
| Geoemydidae | *Heosemys spinosa* | FMNH 260383 | DRS | 2 | SC |
| Geoemydidae | *Mauremys reevesii* | FMNH 6736 | DRS | 2 | SC |
| Geoemydidae | *Melanochelys trijuga* | FMNH 224247 | DRS | 2 | SC |
| Geoemydidae | *Notochelys platynota* | FMNH 224050 | DRS | 2 | SC |
| Geoemydidae | *Orlitia borneensis* | FMNH 224000 | DRS | 2 | SC |
| Geoemydidae | *Pangshura tentoria* | FMNH 259433 | DRS | 2 | SC |
| Geoemydidae | *Sacalia quadriocellata* | FMNH 6605 | ETH | 2 | SC |
| Geoemydidae | *Malayemys subtrijuga* | FMNH 255268 | DRS | 2 | SC |

**Table 1** (*continued*)

| Clade | Species | Catalog Number | Pres | Cat | Acq |
|---|---|---|---|---|---|
| **Geoemydidae** | *Morenia petersi* | FMNH 260377 | DRS | 2 | SC |
| **Geoemydidae** | *Batagur dhongoka* | FMNH 224106 | DRS | 3 | SC |
| **Kinosternidae** | *Claudius angustatus* | FMNH 4165 | DRS | 2 | SC |
| **Kinosternidae** | *Staurotypus triporcatus* | FMNH 4164 | DRS | 2 | SC |
| **Kinosternidae** | *Sternotherus odoratus* | FMNH 83357 | DRS | 2 | SC |
| **Kinosternidae** | *Kinosternon baurii* | FMNH 83436 | DRS | 2 | SC |
| **Pelomedusidae** | *Pelusios sinuatus* | FMNH 12699 | DRS | 1 | SC |
| **Pelomedusidae** | *Pelomedusa subrufa* | FMNH 17173 | DRS | 2 | SC |
| **Platysternidae** | *Platysternon megacephalum* | FMNH 24229 | ETH | 1 | SC |
| **Podocnemididae** | *Podocnemis vogli* | FMNH 73419 | MUM | 2 | SC |
| **Testudinidae** | *Astrochelys radiata* | FMNH 72598 | ETH | 0 | SC |
| **Testudinidae** | *Chelonoidis carbonaria* | FMNH 63916 | DRS | 0 | SC |
| **Testudinidae** | *Chersina angulata* | FMNH 83000 | ETH | 0 | SC |
| **Testudinidae** | *Geochelone elegans* | FMNH 117829 | MUM | 0 | SC |
| **Testudinidae** | *Gopherus polyphemus* | FMNH 83340 | DRS | 0 | SC |
| **Testudinidae** | *Homopus femoralis* | FMNH 17178 | MUM | 0 | SC |
| **Testudinidae** | *Indotestudo elongata* | FMNH 257382 | DRS | 0 | SC |
| **Testudinidae** | *Kinixys belliana* | FMNH 17179 | ETH | 0 | SC |
| **Testudinidae** | *Malacochersus tornieri* | FMNH 252435 | DRS | 0 | SC |
| **Testudinidae** | *Manouria impressa* | FMNH 263045 | DRS | 0 | SC |
| **Testudinidae** | *Psammobates tentorius* | FMNH 17176 | DRS | 0 | SC |
| **Testudinidae** | *Pyxis arachnoides* | FMNH 73308 | ETH | 0 | SC |
| **Testudinidae** | *Stigmochelys pardalis* | FMNH 29277 | DRS | 0 | SC |
| **Testudinidae** | *Testudo graeca* | FMNH 211730 | MUM | 0 | SC |
| **Trionychidae** | *Dogania subplana* | FMNH 241342 | ETH | 3 | SC |
| **Trionychidae** | *Pelodiscus sinensis* | FMNH 24249 | ETH | 3 | SC |
| **Trionychidae** | *Rafetus euphraticus* | FMNH 19492 | ETH | 3 | SC |
| **Trionychidae** | *Apalone mutica* | FMNH 7845 | ETH | 3 | SC |
| **Trionychidae** | *Lissemys punctata* | FMNH 73919 | ETH | 3 | SC |
| — | *Proganochelys quenstedtii* | SMNS 16980 | cast | ? | PH |
| — | *Proterochersis robusta* | SMNS 17561 | fossil | ? | PH |
| **Thalassochelydia** | *Plesiochelys bigleri* | MJSN CBE-0002 | fossil | ? | SC |

**Notes.**

Abbreviations: DRS, dry skeletal specimen; ETH, complete specimen preserved in ethanol; MUM, complete mummified specimen; SC, 3D Scanner; PH, Photogrammetry reconstruction.

and posterior-most points along the midline of the carapace (landmarks 1 and 2) and plastron (landmarks 3 and 4), the anterior and posterior limits of the contact of the axillary (landmarks 5 and 8) and inguinal buttress (landmarks 6 and 7) with the peripheral series, and the median point between the buttresses, typically the hyo/hypoplastral contact with the peripheral series (landmarks 9 and 10). These primary landmarks define the start and end points of the twelve semi-landmark curves (Fig. 1), in particular the outline of the carapace (curves C1 and C2), the doming of the carapace (curves C3 and C12), the midline and cross section of the plastron (curves C4 and C11), the outline of the anterior
**Table 2  Description of the four different sub-dataset used in the analyses.** The listed landmarks and semilandmarks are shown in Fig. 2.

|  | Description | Landmarks | SM |
|---|---|---|---|
| SET1 | All landmark data is included. | all | all |
| SET2 | The outline of the of the carapace | 1, 2 | C1, C2 |
| SET3 | The transverse cross-section of the shell | 9, 10 | C11, C12 |
| SET4 | The longitudinal cross-section of the shell | 1, 2, 3 ,4 | C3, C4 |

Notes.
   Abbreviations: SM, semilandmarks.

and posterior plastral lobes (curves C5, C7, C8, C10), and the bridge (i.e., contact of the plastron with the carapace, curves C6 and C9).

Landmarks were set directly onto the 3D models using the software *Checkpoint* (Stratovan). The curves were captured in a two-step process. For the first step, semilandmarks were manually set along the curves of the specimen using the "curve" function of *Checkpoint*. The resulting curves are not yet comparable to one another, as they utilize a different number of unevenly set semilandmarks. The primary semilandmarks curves were therefore resampled in *R* v3.6.3 (*R Core Team, 2020*) to produce an equidistant repartition of 255 points along the curves (*Gunz & Mitteroecker, 2013*) using the *digit.curves* function of the package *geomorph* v3.2.1 (*Adams & Otárola-Castillo, 2013*; *Adams, Collyer & Kaliontzopoulou, 2020*).

In order to discuss which components provide the most variation and identify which parameters of the shell represent the best proxy to infer the ecology of turtles, we produced four datasets with different landmarks and semilandmarks configurations (Table 2, Fig. 2) capturing several aspects of the shell. SET 1 utilizes all landmarks, SET 2 the perimeter of the carapace, SET 3 the transverse cross-section, a proxy for doming, and SET 4 the cross-section, a proxy for the hydrodynamics of the shell.

## Classification of habitat preferences

In order to investigate the relationships between habitat preferences and shell shape among the extant turtles in the sample, it is necessary to classify them by their ecology (Table 1). As gradual variation is apparent between habitat categories, it is difficult to implement this step, we used the method of *Foth et al. (2019)*, which categorizes turtles by the development of the webbing of their forelimbs as an ecological proxy (Table 3, Fig. 3). This is based on the justifiable assumption that the degree of webbing correlates with the amount of time the turtle spends in water. In contrast to defining ecological categories based on imprecise descriptions from the literature (e.g., "terrestrial," "poorly aquatic", "semi aquatic" or "fully aquatic"), this approach is more objective, as webbing can be easily observed in museum specimens or the scientific literature (e.g., *Ernst & Barbour, 1989*; *Bonin, Devaux & Dupré, 1998*). Our five primary categories include "no webbing" (0), "poorly webbed" (1), "fully webbed," with webbing reaching the base of the claws (2), "extensive webbing," with at least one claw being enveloped (3), and "flippers" (4). The scoring for each species is provided in Table 1. We also tested an alternative classification, which is a combination of the previously described categories, defined as "terrestrial" (including category 0, "not
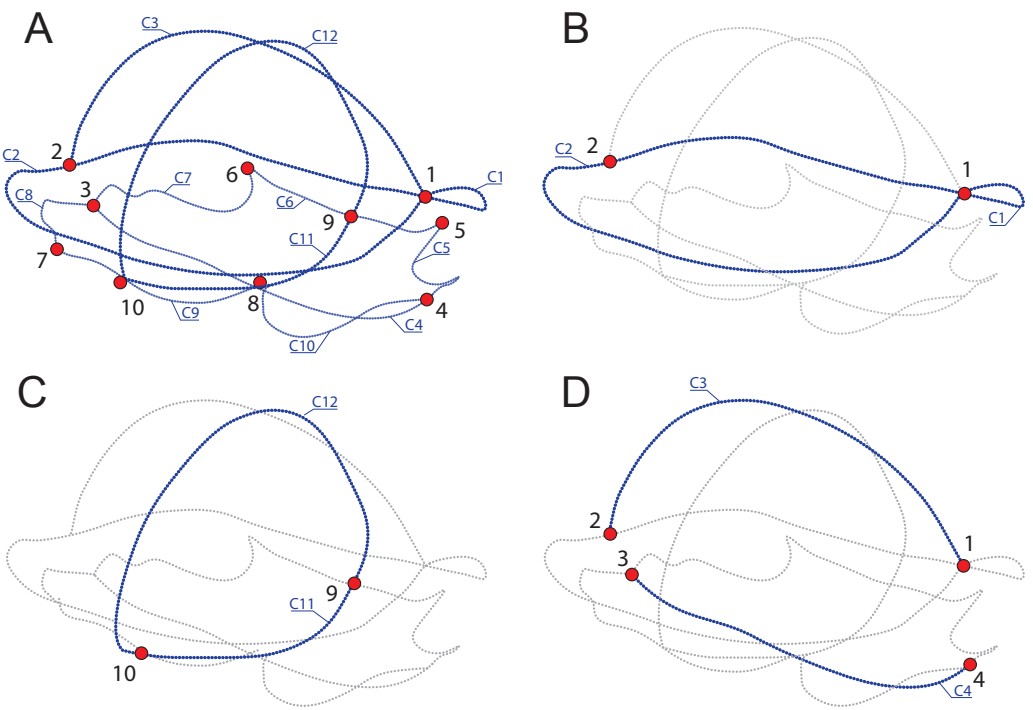

**Figure 2 Subsamples used in this study.** (A) SET1, all landmarks and semilandmarks combined. (B) SET2, outline of the carapace only. (C) SET3, transverse cross-section only. (D) SET4, longitudinal cross-section only. Landmarks are numbered from 1 to 10. Curves composed of semilandmarks are numerated from C1 to C12.

webbed"), "semi-aquatic" (including category 1 and category 2, "poorly webbed" and "fully webbed") and "aquatic" (including category 3 and category 4, "extensive webbing" and "flippers").

## Analyses of morphometric data

In order to compare the shapes of the turtle shells we obtained, all sets of landmarks were scaled, translated, and rotated using Generalized Procrustes superimposition (GPA: *Rohlf & Slice, 1990*). This procedure was undertaken in *R* using the function *gpagen* in *geomorph*. The semilandmarks were slid using bending energy (*Gunz, Mitteroecker & Bookstein, 2005*).

To test for the impact of allometric shape variation we used the log-transformed centroid size of the specimens of each dataset and produced a linear regression against Procrustes shape (see *Drake & Klingenberg, 2008*). The regression was computed using the function *procD.lm* in the *R* package *geomorph*. The ANOVA (analysis of variance) was performed with 1,000 permutations.

Then, we performed a Principal Component Analysis (PCA), which is a commonly used method to convert a set of data into a set of independent variables. The PCA was computed using the function *gm.prcomp* in the *R* package *geomorph*.

We first tested for a correlation between ecology and shell shape using a linear discriminant analysis (LDA), which distinguishes morphological differences between

**Table 3** Description of ecological categories used in this study based on the webbing of the forelimb as a proxy.

| Cat. | Webbing type |
| --- | --- |
| Cat.0 | Webbing absent. This morphology is associated with an exclusively terrestrial ecology (Fig. 3A). |
| Cat.1 | Minor webbing present between the first phalanges of all fingers (Fig. 3B). This morphology is typical for turtles that spend a moderate amount of time in water. |
| Cat.2 | Extensive webbing present that reaches the ungual phalanx of all digits (Fig. 3C). The associated ecology is semi-aquatic to aquatic in behavior. This is the largest category, including turtles that inhabit lakes, rivers, and ponds and that either swim actively or walk at the bottom. |
| Cat.3 | Extensive webbing present that envelopes at least one digit completely, typically digit V (Fig. 3D). This category is typical for highly aquatic turtles that rarely leave the water, including several riverine testudinoids and all trionychids. |
| Cat.4 | The forelimb is elongated, the webbing is extensive, and the limb shaped into a soft flipper or hard paddle (Fig. 3E). This category is represented by marine cheloniids and freshwater aquatic carettochelyids. |

**Notes.**
Abbreviations: Cat, category.

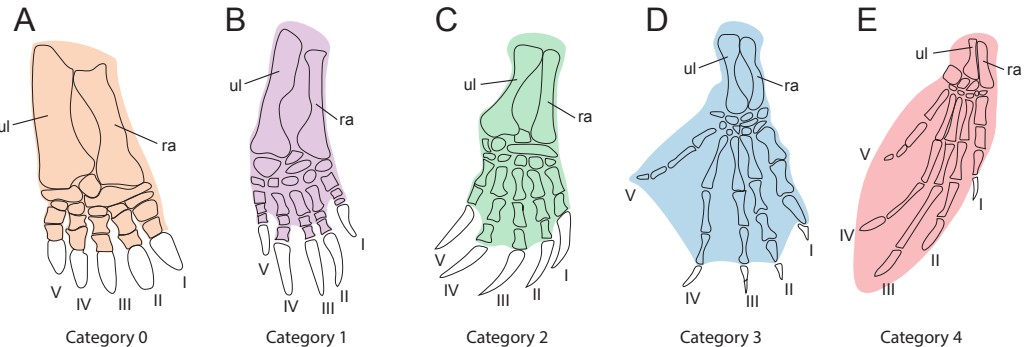

**Figure 3** Webbing types of the forehand used for ecological classification. (A) Webbing absent. (B) Poorly webbed, webbing only present at the base of the digits. (C) Fully webbed, webbing reaches the base of the claws. (D) Webbing extensive, webbing envelopes at least one claw. (E) Manus modified into elongate flipper or paddle. Digits are numbered from 1 to 5 using Roman numerals. Abbreviations: ul, ulna; ra, radius.

groups (*Fisher, 1936*; *McLachlan, 2004*). LDA identifies the axes that maximize the separation between multiple classes, in our case the ecological categories we select. LDA is based on those principal components (PC) that contain significant shape information. The number of significant PC scores kept was estimated using the broken-stick method (*Frontier, 1976*; *De Vita, 1979*; *Jackson, 1993*, see Fig. S1). The LDA tested the performance of an *a priori* classification model and assigned specimens of unknown ecology to a specific category. The LDA was performed using the function *lda* from the package *MASS*

(*Ripley et al., 2013*) and was used for the calculations. To test the accuracy of the predictions and prevent overfitting, we performed the analysis with and without leave-one-out cross-validation.

Furthermore, we also performed a phylogenetic flexible discriminant analysis (pFDA). In contrast to LDA, pFDA addresses the impact of phylogeny on the data to provide predictions (*Motani & Schmitz, 2011*). The phylogenetic tree used for the pFDA is based on *Pereira et al. (2017)*, which is the best sampled molecular tree available for extant turtles. The original tree, which consists of 294 extant turtles, was pruned to only include the taxa present in the sample and then time-calibrated based on *Joyce, Schoch & Lyson (2013)*. The extinct turtles *Proganochelys quenstedtii* and *Proterochersis robusta* were then added as stem-turtles following *Joyce (2007)*, with *Proganochelys quenstedtii* as the most basal turtle in the tree. *Plesiochelys bigleri* was placed as sister group to Cryptodira following *Anquetin, Püntener & Joyce (2017*; Fig. 4). The ages for the time calibration of the fossil taxa was taken from *Joyce (2017)* and *Anquetin, Püntener & Joyce (2017)*. Alternative positions for these taxa can be found, among others, in *Szczygielski & Sulej (2016)* or *Evers & Benson (2019)*. The strength of the phylogenetic signal is estimated by the Pagel's lambda-value ($\lambda$), which varies from 0 to 1, with 0 denoting the lack of a phylogenetic signal and 1 denoting a strong phylogenetic signal under Brownian motion (*Pagel, 1999*). This corrects for the phylogenetic bias that can occur in the dataset. The discriminant analysis hereby attempts to predict the ecology of each data point based on the input data. This step produces the confusion matrix that summarize the results. The R code used for the pFDA was originally published by *Motani & Schmitz (2011)*, which in return was adapted from *Hastie, Tibshirani & Buja (1994)*. The code was adapted for the purpose of this study.

## RESULTS

### Allometry

The results of the linear regression and the ANOVA indicate no correlation between shape and log-transformed centroid size ($R^2 = 0.0235$, $P$-value = 0.134; Fig. 5), indicating the absence of an interspecific allometric signal. We therefore, did not calculate the non-allometric residuals of the Procrustes coordinates.

### Principal Component Analysis (PCA)

For SET1 (Fig. 6A), PC1 explains 28.81% of the total shape variation. Most of the variation pertains to the height of the dome of the shell and the relative size of the plastron, in that highly domed shells have enlarged plastra (negative PC scores) and flattened shells have small plastra (positive PC scores). Surprisingly, turtles categorized by the presence of flippers (category 4) are scattered across the plot. PC2 explains 14.5% of the variation. Like PC1, it pertains to the height of the dome and the relative size of the plastron, in that highly domed shells have a small plastron (negative PC scores) and flattened shells possess an enlarged plastron (positive PC scores). The PCA plot for SET1 shows an overlap of most ecological categories. *Proterochersis robusta* groups with non-webbed (category 0) and poorly-webbed (category 1) turtles with domed-shells, while *Proganochelys quenstedtii* and *Plesiochelys bigleri* are closer to turtles with flattened shells.

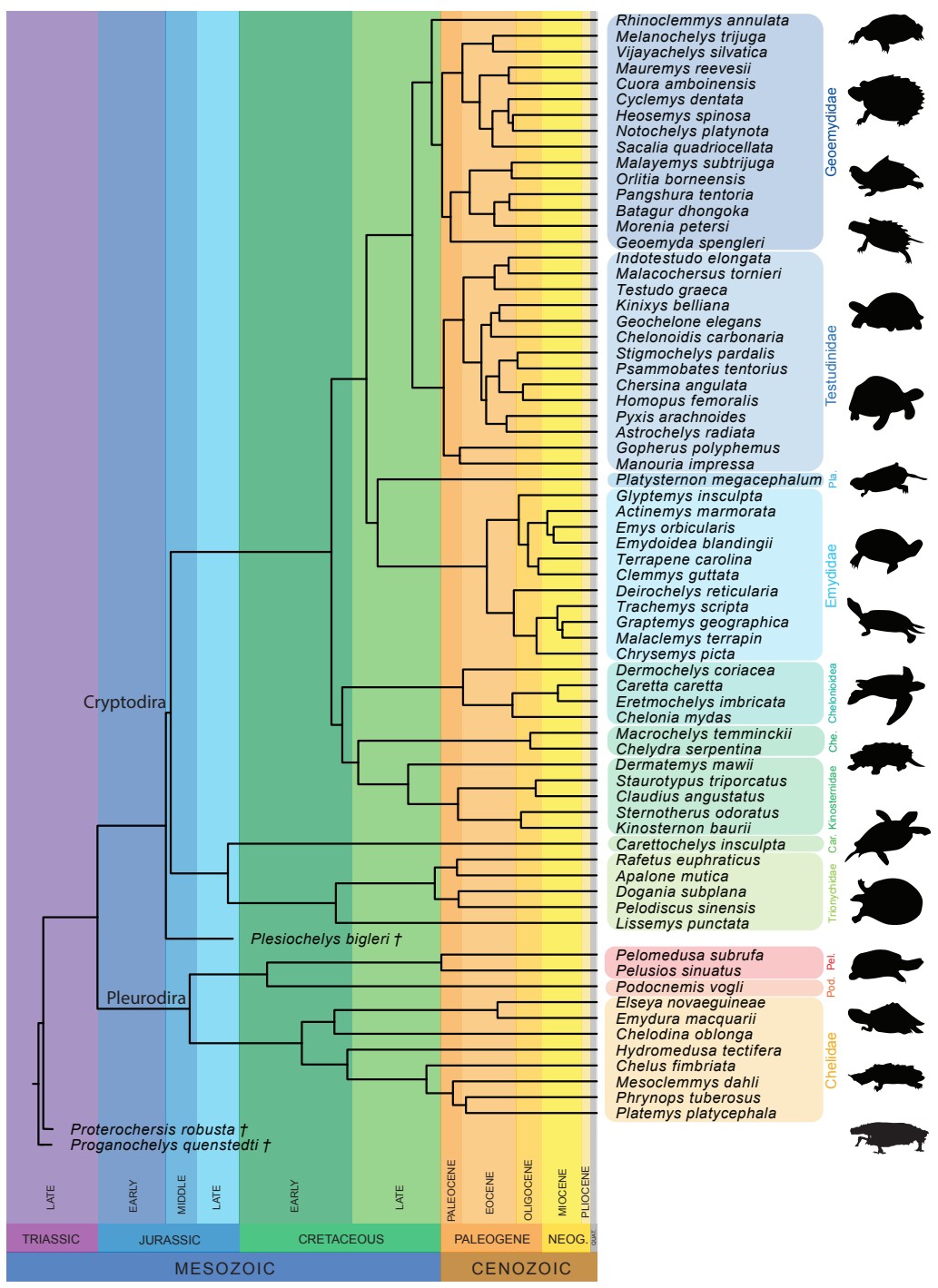

**Figure 4** **Time-calibrated phylogeny of 72 species used in the study based on *Pereira et al. (2017)*.** Abbreviation: Car, Carettochelyidae; Che, Chelydridae; Pel, Pelomedusidae; Pla, Platysternidae; Pod, Podocnemididae.

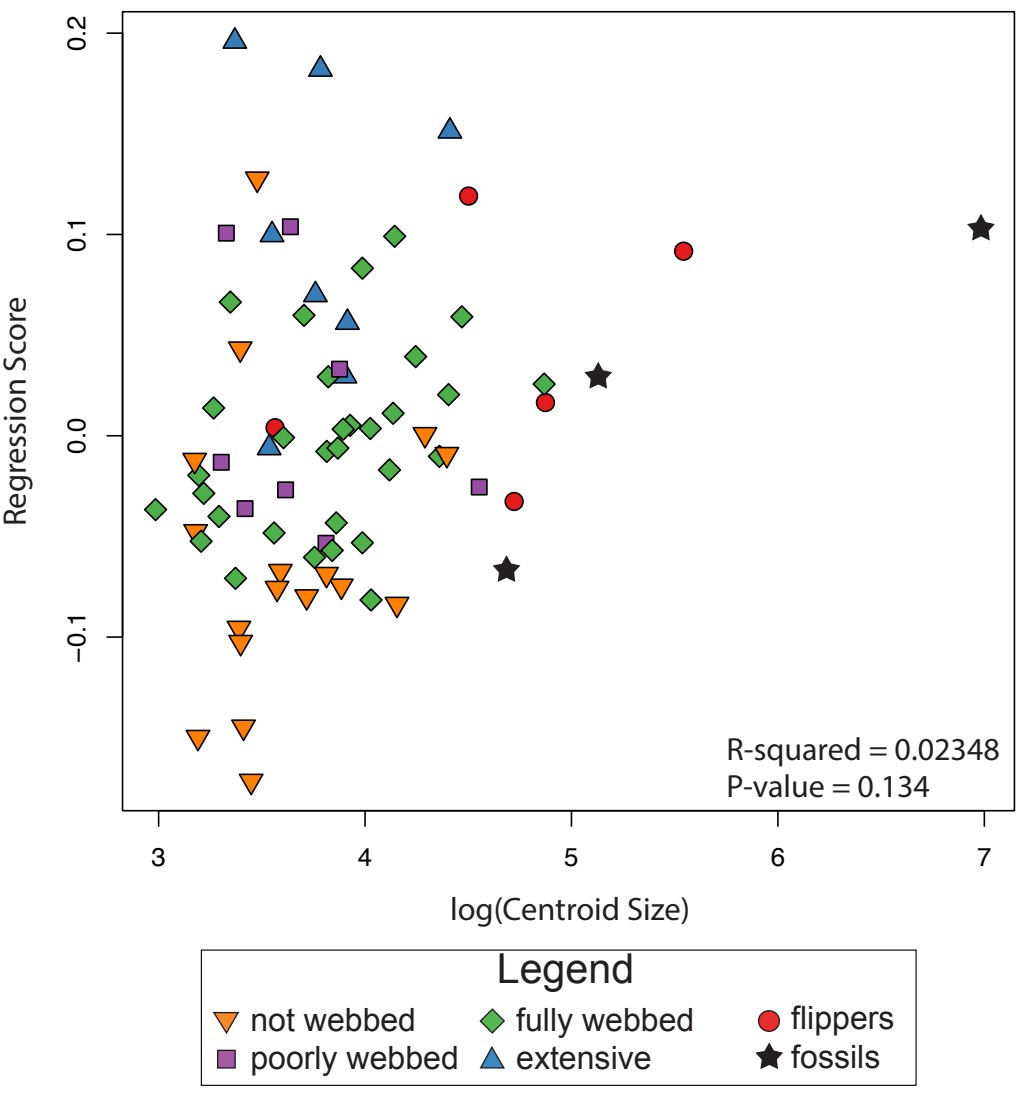

**Figure 5  Relationship between size and shape.** The graph shows regression scores (shape) plotted against Log(CSize) to highlight possible allometric correlations.

SET2, which describes the outline of the carapace (Fig. 6B), PC1 explains 37.41% of the total variation. The shape of the outline of the shell varies from elongate (negative scores) to rounded, being almost as wide as long (positive scores). PC2 explains 19.25% of the total variation and captures shell width from broad (negative PC scores) to narrow (positive scores). Turtles with flippers (category 4) plot closely together but are still nested with the group of fully webbed turtles (category 2). The included fossils do not group with any particular category. In addition, the fossils tend towards positive PC1 scores, in the left part of the graph, which corresponds to a more rounded morphology.

PC1 of SET 3, which captures the transverse cross-sectional shape of the shell, explains 68.44% of the total variance, most of which pertains to the height of the dome, from flat (negative scores) to highly domed (positive scores) (Fig. 6C). PC2 carries 16.71% of the

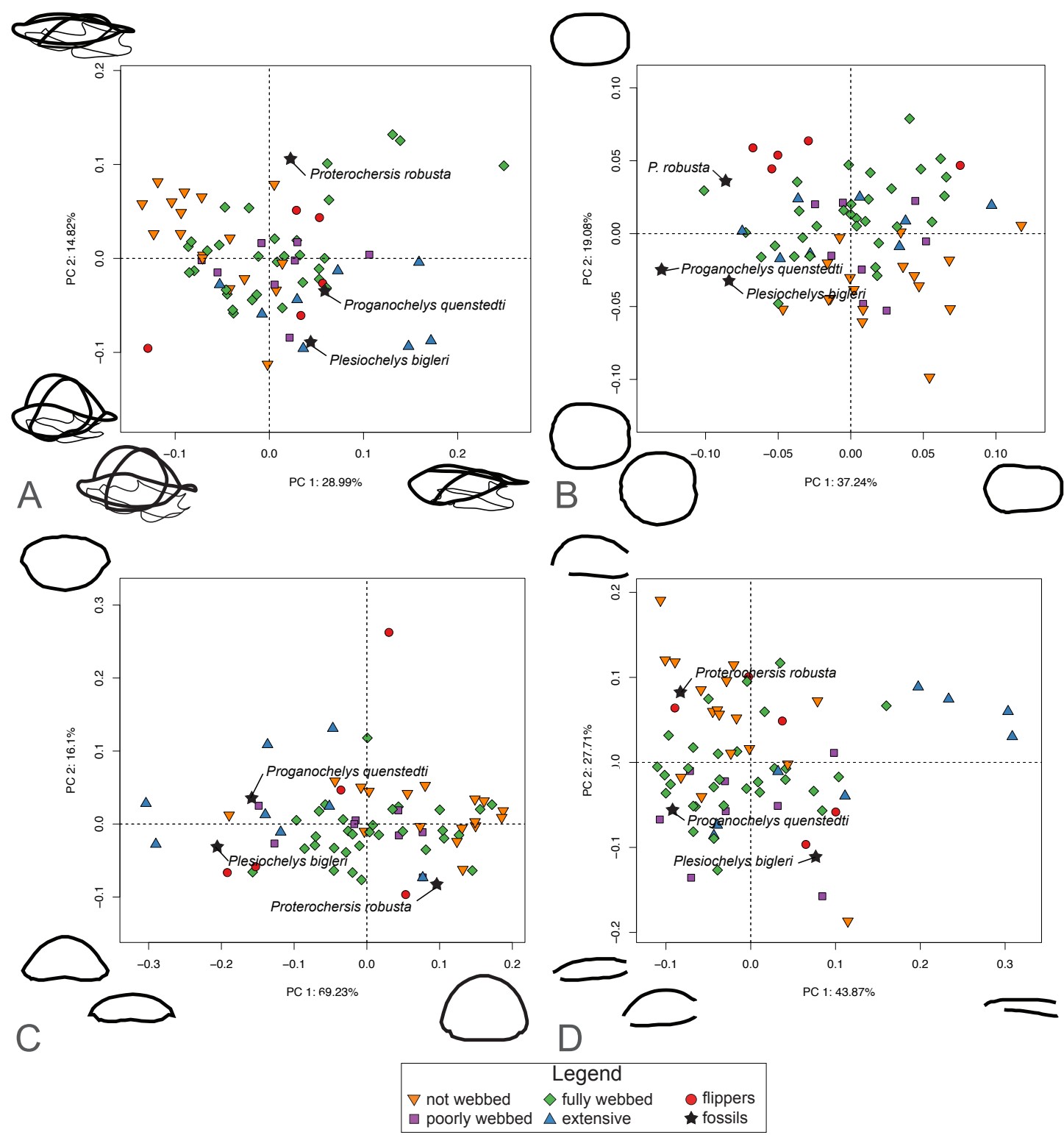

**Figure 6** **Results of the PCA based on four different dataset configurations.** (A) All landmarks and semilandmarks curves, SET1. (B) Outline of the carapace, SET2. (C) Transverse cross-section, SET3. (D) Longitudinal cross-section, SET4.

total variance and mostly explains the cross-section of the shell from domed carapaces with a flat plastron (negative scores) to flattened carapaces with a convex plastron (positive scores). As with the previous SETs, the ecological categories strongly overlap each other. *Proganochelys quenstedtii* and *Plesiochelys bigleri* plot on the negative site of PC1, while *Proterochersis robusta* is found on the opposite of PC1. Part of the overlap is explained by the presence of the outliers for various categories, in particular the pancake tortoise (*Malacochersus tornieri*), which is a greatly flattened terrestrial turtle, or the leatherback turtle (*Dermochelys coriacea*), which is a marine turtle with a strongly convex plastron.

SET4 investigates shape variation to the longitudinal cross-section of the shell (Fig. 6D). PC1 explains 44.4% of the total variance. Turtles represented by negative scores have a domed morphology and a long plastron, in which the dome is accentuated in the anterior part of the shell. Turtles represented by positive scores capture flattened carapaces with short plastra. Here, the carapace overhangs the posterior end of the plastron. PC2 represents 27.1% of the total variance. Negative scores correspond to a flat-shaped carapace and elongated plastron. Positive scores describe a domed carapace, with the maximum curvature in the posterior section of the shell that overhangs the plastron. As with the other SETs, the PCA shows a big overlap in the distribution of various ecological categories. Trionychids nevertheless plot closely together in the positive part of PC1 scores. *Proterochersis robusta* plots close to the terrestrial turtles (category 0), while *Proganochelys quenstedtii* plots in the "fully webbed" range (category 2). *Plesiochelys bigleri* plots towards the left of the graph (see Table S3).

## Linear discriminant analysis results

The recognition of the ecological categories by the confusion algorithm for the linear discriminant analysis (LDA) is variable depending on the subset (SET) used (Table 4, detailed tables are provided in Table S4). The main error is in a range between 25% and 40% of misclassification for each SET. However, SET1 (25.3% of misclassification) gives the best results as compared to the other SETs. In fact, in SET1, all categories are recognized at least at a rate of 50%. In SET2 and SET4, species defined as "poorly webbed" (category 1) are not well identified (38%). For the SET3, which represents the transverse cross-section, the categories flippers (category 4, 60%, while 100% recognized for all the other SETs) and poorly webbed (category 1, 13%) are poorly distinguished. The outcome of the confusion matrix gives the most robust results for SET1, among all the arrangements. The use of all data is therefore better than the use of just one component. After cross-validation, the total error of correct identification increased moderately for SET1 (32%), SET2 (36%) and SET4 (37%, see Table S4 for all confusion matrices). While all categories are still recognized at a rate of minimum 50% for SET1, recognition of "poorly webbed" turtles (category 1) and "extensive webbing" (category 3) drop significantly for SET2 (38% and 25%) and SET3 (38% and 0%). There is also a drop in the recognition for turtles having flippers for SET3 (40%). On the other hand, "not webbed" turtles (category 0) and "fully webbed" turtles (category 2) stayed highly stable. These results indicate overfitting of the training data, indicating that the predictions are partially dependent on sample-size. However, the

**Table 4  Confusion matrix showing the recognition of ecological category per SET in the LDA.** Each line of the table describes the results for each of the four sub-analyses (SET1 to SET4). Each column corresponds to a webbing category. All results are expressed in percent. The last column of the table provides the main error in percent.

|  | Cat 4 | Cat 3 | Cat 2 | Cat 1 | Cat 0 | Error |
|---|---|---|---|---|---|---|
| **SET1** | 100 | 50 | 90 | 63 | 70 | 25.31 |
| **SET2** | 100 | 50 | 90 | 38 | 94 | 25.61 |
| **SET3** | 60 | 50 | 87 | 13 | 88 | 40.43 |
| **SET4** | 100 | 63 | 84 | 38 | 76 | 27.93 |

outcome of the confusion matrix using cross-validation still reveals that SET1 performs better than other configurations.

For SET1 (Fig. 7A), three groups of extant turtles are discriminated: (1) turtles lacking webbing (category 0); (2) turtles ranging from non-webbed to fully webbed turtles (category 0–2); (3) and turtles with extensive webbing (category 3) and flippers (category 4). For SET2 (Fig. 7B), which corresponds to the outline of the carapace, only turtles with flippers (category 4) are well-discriminated, as these taxa all possess a distinctive tear-drop-shaped shell (see mean shapes per category in Fig. S2). For SET3 and SET4 (Fig. 7C, Fig. 7D), the webbing categories greatly overlap each other. There is a gap between the two extreme categories (not webbed and flippers) but no category is discriminated. There is a trend along the LD1, with terrestrial adaptations (i.e., no or minor webbing) on the negative side, and aquatic adaptations (i.e., extensive webbing or flippers) on the positive side.

The predictions of the webbing (and thus ecology) of the fossil turtles are variable between the SETs (see Fig. 7; Table 5). For SET1, all fossil turtles are identified as having "fully webbed" forelimbs (category 2). However, *Plesiochelys bigleri* plots just at the limits between "fully webbed" (category 2) and "extensively webbed" and "flipper-shaped" forehand (category 3 and 4) and *Proterochersis robusta* at the limit between "poorly webbed" (category 1) and "fully webbed" (category 2) turtles (Fig. 7A). *Proganochelys quenstedtii* plots within the "fully webbed" (category 2) turtles. For SET2, the fossil turtles are identified as either fully webbed (category 2) or poorly webbed (category 1), but plot further away from the extant groups, except for *Plesiochelys bigleri*, which groups with fully webbed (category 2) turtles but was determined to be "poorly webbed" with a probability of 49% (see Table 5). For SET3, *Proterochersis robusta* is predicted to be "fully webbed" (category 2), but only with a probability of 38%. On the other hand, *Plesiochelys bigleri* is predicted to have been "extensively webbed" (category 3) with a low probability of 49% while *Proganochelys quenstedtii* groups with turtles that are "poorly webbed" (category 1), also with a low probability (47%). Finally, for SET4, *Proganochelys quenstedtii* is predicted to have been "poorly webbed" (category 1), while *Proterochersis robusta* and *Plesiochelys bigleri* are reconstructed as "fully webbed" (category 2), which is consistent with what can be observed on the graph.

## Phylogenetic flexible discriminant analysis results

The confusion matrix based on the phylogenetic flexible discriminant analysis (pFDA) shows good recognition of ecological variables (expressed by the degree of webbing in the

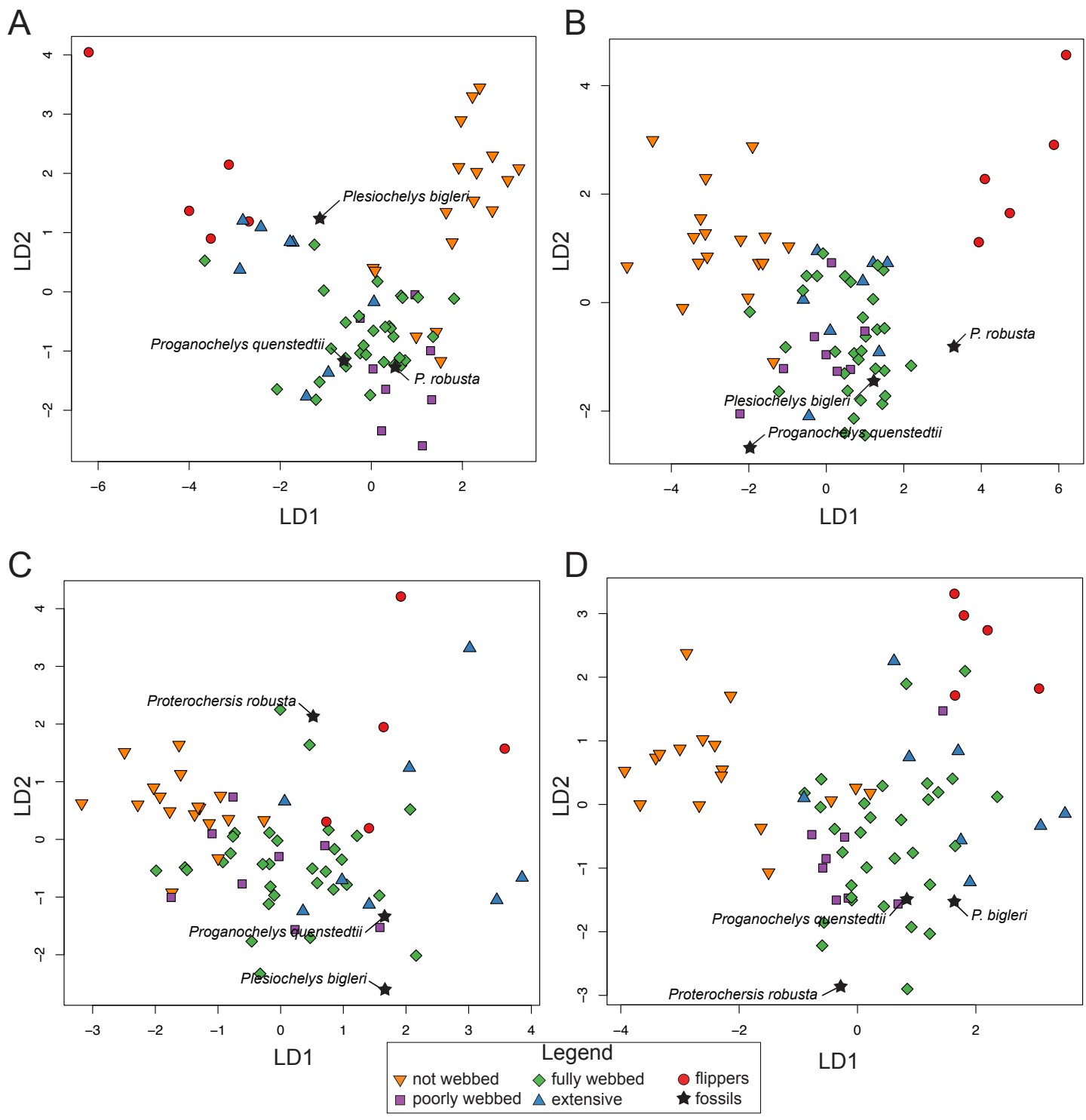

**Figure 7** **Results of the discriminant analysis (LDA) based on four different dataset configurations.** (A) All landmarks and curves, SET1. (B) Outline of the carapace, SET2. (C) Transverse cross-section, SET3. (D) Longitudinal cross-section, SET4. All data are available in the Table S4.

forelimbs) for extant species (Table 6, detailed tables are provided in Table S4). The analysis including all landmarks and curves (SET1) shows consistent results between 50 to 100%

**Table 5  Prediction matrix for the fossils included in the study based on four different dataset configurations based on the linear discriminant analysis (LDA).** Complete data are available in the Table S6.

|  | SPECIES | PREDICTION | CAT. | PROB. |
|---|---|---|---|---|
| SET1 | *Plesiochelys bigleri* | fully webbed | 2 | 0.95 |
|  | *Proterochersis robusta* | fully webbed | 2 | 0.98 |
|  | *Proganochelys quenstedtii* | fully webbed | 2 | 0.99 |
| SET2 | *Plesiochelys bigleri* | poorly webbed | 1 | 0.57 |
|  | *Proterochersis robusta* | fully webbed | 2 | 0.98 |
|  | *Proganochelys quenstedtii* | fully webbed | 2 | 0.69 |
| SET3 | *Plesiochelys bigleri* | extensive webbing | 3 | 0.49 |
|  | *Proterochersis robusta* | fully webbed | 2 | 0.38 |
|  | *Proganochelys quenstedtii* | fully webbed | 2 | 0.47 |
| SET4 | *Plesiochelys bigleri* | fully webbed | 2 | 0.79 |
|  | *Proterochersis robusta* | fully webbed | 2 | 0.88 |
|  | *Proganochelys quenstedtii* | poorly webbed | 1 | 0.93 |

**Table 6  Confusion matrix showing the recognition of ecological category per SET in the pFDA.** Each line of the table describes the results for each of the four subsets (SET1 to SET4). Each column corresponds to a webbing category. All results are presented in percent. The last column of the table provides the main error in percent.

|  | Cat 4 | Cat 3 | Cat 2 | Cat 1 | Cat 0 | Error |
|---|---|---|---|---|---|---|
| **SET1** | 100 | 50 | 94 | 63 | 70 | 24.67 |
| **SET2** | 100 | 63 | 94 | 38 | 94 | 22.47 |
| **SET3** | 60 | 50 | 87 | 13 | 76 | 42.79 |
| **SET4** | 100 | 50 | 83 | 38 | 76 | 30.43 |

accuracy for the different webbing categories. SET2, which describes the outline of the carapace is slightly better regarding the correct identification of most webbing categories, except for minor webbing (category 1). SET3 and SET4, however, fail to identify turtles with minor webbing (category 1) and extensive webbing (category 3). The outcome in the confusion matrix gives the most solid results for SET1 among all arrangements. Therefore, higher accuracy is gained when using all landmarks and semilandmarks in combination with phylogeny (Table 6).

The pFDA results for extant turtles are similar to the LDA results for SET1 and SET2. The distribution, however, is variable for SET3 and SET4. In SET1, the graph is divided into three major groups. One is composed of turtles with not-webbed morphologies, one includes turtles with poorly webbed and fully webbed forelimbs, and a last one with turtles having extensive webbing and flipper-shaped forelimbs (Fig. 8A). In SET2 only turtles with flippers are well discriminated. The results for SET3 and SET4 show much overlap between all categories. The predictions for fossils are not congruent depending on the arrangement being used. All fossil turtles are predicted to be "minor-webbed" (category 1) to "flipper-shaped" (category 4), which suggests aquatic habitat preferences. However, there is great variability in the predictions depending on the configuration of the dataset (SET1 to SET4). Although *Plesiochelys bigleri* is resolved as having flippers

**Table 7** Prediction matrix for the fossils included in the study based on four different dataset configurations based on the phylogenetic flexible discriminant analysis (pFDA). Complete data are available in Table S7.

|  | SPECIES | PREDICTION | CAT. | PROB. |
|---|---|---|---|---|
| SET1 | *Plesiochelys bigleri* | flippers | 4 | 0.87 |
|  | *Proterochersis robusta* | fully webbed | 2 | 0.99 |
|  | *Proganochelys quenstedtii* | fully webbed | 2 | NaN |
| SET2 | *Plesiochelys bigleri* | poorly webbed | 1 | 0.91 |
|  | *Proterochersis robusta* | flippers | 4 | 0.99 |
|  | *Proganochelys quenstedtii* | poorly webbed | 1 | NaN |
| SET3 | *Plesiochelys bigleri* | fully webbed | 2 | 0.71 |
|  | *Proterochersis robusta* | flippers | 4 | 0.99 |
|  | *Proganochelys quenstedtii* | extensive webbing | 3 | 0.99 |
| SET4 | *Plesiochelys bigleri* | fully webbed | 2 | 0.56 |
|  | *Proterochersis robusta* | fully webbed | 2 | 0.99 |
|  | *Proganochelys quenstedtii* | poorly webbed | 1 | NaN |

(category 4) and plots with extant turtles for SET1, the Triassic fossil turtles are resolved as "fully webbed" (2) but plot further away from the extant group. *Proganochelys quenstedtii* and *Proterochersis robusta* do not group close together with any other turtle. For SET2, *Plesiochelys bigleri* is grouped again within the extant group, contrary to *Proganochelys quenstedtii* and *Proterochersis robusta*, which are found to be outliers. *Plesiochelys bigleri* is predicted as poorly webbed" (category 1), while *Proterochersis robusta* and *Proganochelys quenstedtii* are predicted to have "flippers" (category 4) and "poorly webbed" (category 1). In SET3, all fossils plot outside of the extant groups, even if the algorithm gives predictions such as extensive webbing (category 3) for *Proganochelys quenstedtii*, "fully webbed" (category 2) for *Plesiochelys bigleri*, and "flippers" (4) for *Proterochersis robusta* (Table 7). For the SET4, the fossils plot again outside of the extant categories and are predicted to be "fully webbed" (category 2) for *Proterochersis robusta* and *Plesiochelys bigleri* and as "poorly webbed" (category 1) for *Proganochelys quenstedtii*.

## Ecological categories

It is notable that the categories poorly webbed (category 1) and fully webbed (category 2) overlap each other in both LDA and pFDA, just as the categories extensive webbing (category 3) and flippered (category 4). However, the pFDA is not very insightful concerning the webbing/ecology of fossil turtles. To investigate the impact of the categorization done herein, the LDA analysis was performed on the SET1 again using a different combination of categories. In particular, the five previously used categories were reclassified for this purpose into three novel categories, herein defined as "terrestrial" (including category 0, not webbed), "semi-aquatic" (including category 1 and category 2, poorly webbed and fully webbed) and "aquatic" (including category 3 and category 4, extensive webbing and flippers). The results of this secondary analysis are provided in the update confusion table (Table 8) and graphs (Fig. 9B).

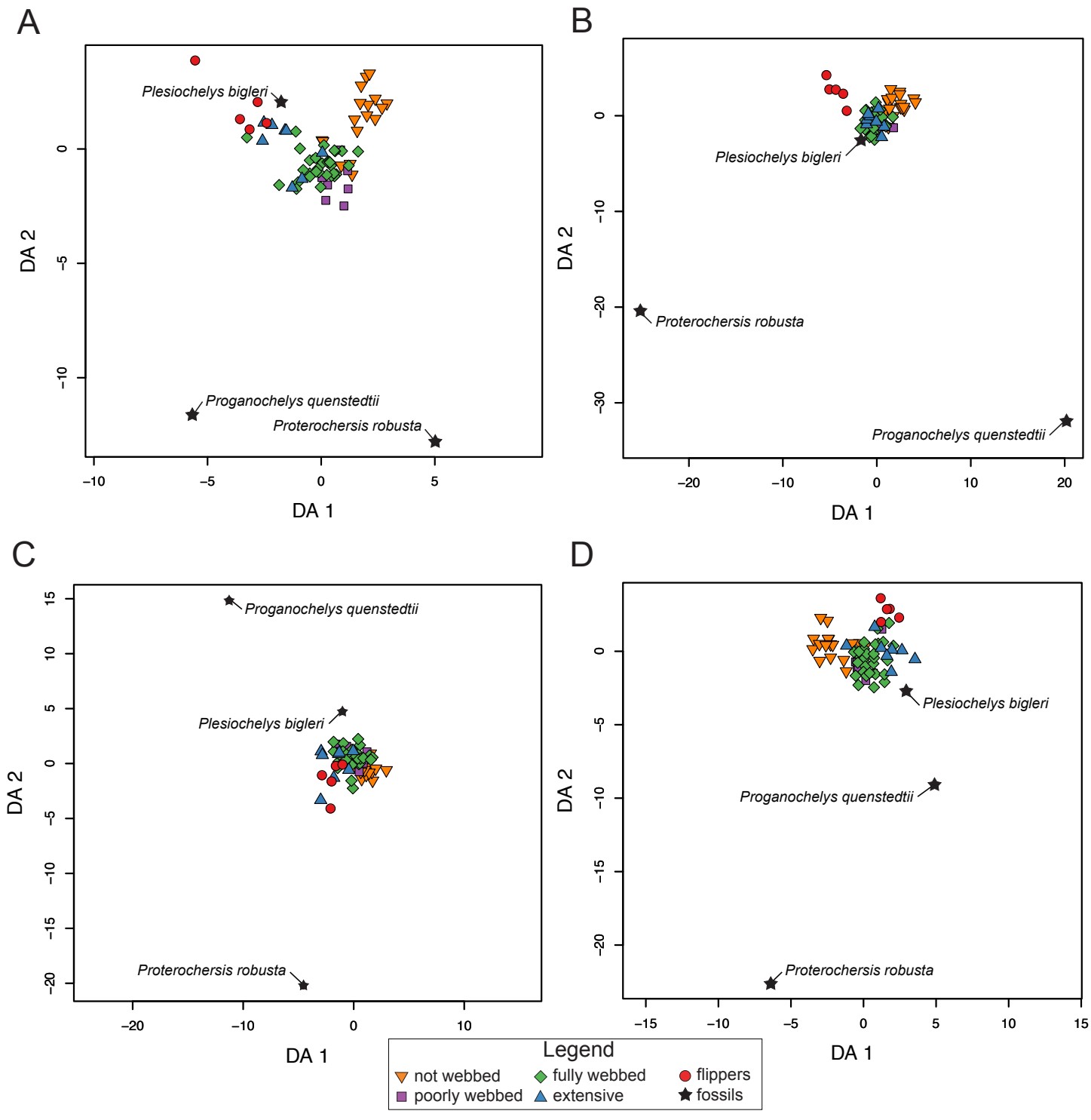

**Figure 8   Results of the phylogenetic flexible discriminant analysis (pFDA) based on four different dataset configurations.** (A) All landmarks and curves, SET1. (B) Outline of the carapace, SET2. (C) Transverse cross-section, SET3. (D) Longitudinal cross-section, SET4. Complete data are available in Table S5.

**Table 8  Confusion matrix for the LDA with only three ecological categories applied to SET1 (Misclassification Rate: 14%).**

|  | AQ | SA | TR |
| --- | --- | --- | --- |
| AQ | 10 | 3 | 0 |
| SA | 1 | 38 | 0 |
| TR | 0 | 5 | 12 |

**Notes.**

Abbreviations: AQ, aquatic (flippers and extensive webbing); SA, semi-aquatic (poorly webbed and fully webbed); TR, terrestrial (not webbed).

Rows represent the predictions and columns represent the true ecology.

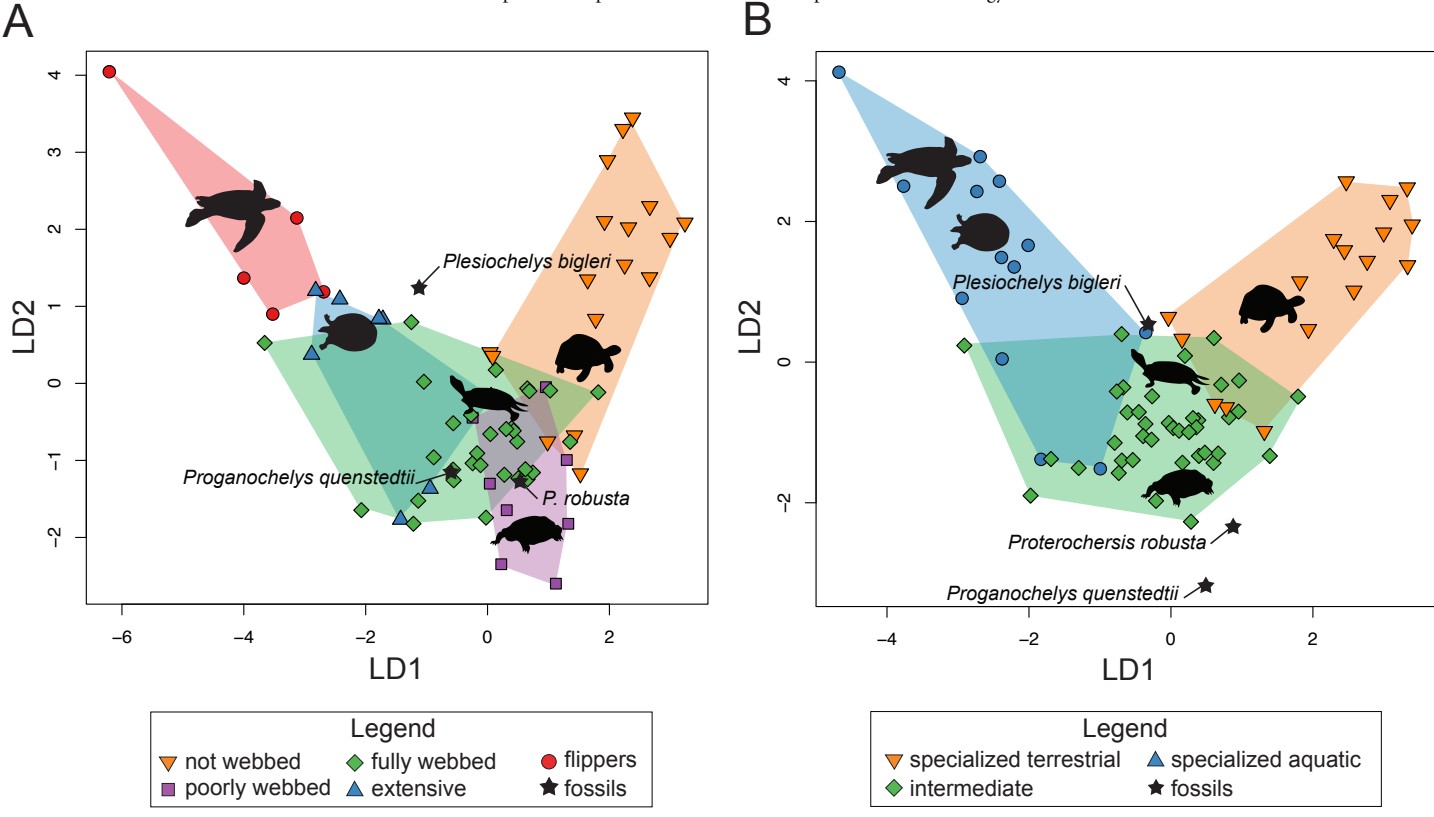

**Figure 9  Comparative results of the linear discriminant analysis (LDA) including predictions for fossil species.** (A) Analysis using five ecological categories. (B) Analysis using three ecological categories.

The misclassification rate for the confusion matrix associated with the three new categories (18.4%) is lower than what is observed in the one with five categories (25.3%). For instance, semi-aquatic turtles are well recognized (38 of 39), but some aquatic (3 over 13) and terrestrial turtles (5 over 17) are still misclassified. However, in general, the dataset containing three categories (Fig. 9B) gives similar results when compared with the original dataset defined by five categories (Fig. 9A). Both groupings show no overlap between the terrestrial and the aquatic categories (see Fig. S3). However, the third category of semi-aquatic turtles remains poorly discriminated. When it comes to fossils specimens, the results are similar between the two grouping classifications (Table 9). *Plesiochelys bigleri* falls between fully webbed (2) and extensive webbing (3) in the first classification

model (Fig. 9A) and remains at this position in the second plot (Fig. 9B), between the semi-aquatic and the aquatic category. Moreover, in the model with three categories, *Proganochelys quenstedtii* and *Proterochersis robusta* plot further away from the extant groups. It appears that splitting the semi-aquatic category into two (poorly-webbed and fully-webbed) gives a more precise placement for the Triassic turtles such as they plot closer to the extant groups, even if there is poor discrimination between these two categories.

## DISCUSSION

### Results for extant turtles

In order to determine the paleoecology of extinct species, paleontologists often draw from correlations found among the shape and ecology of extant organisms (e.g., *Cassini, 2013*) for ungulates; (*Cooke, 2011*) for platyrrhine primates; (*Forrest, Plummer & Raaum, 2018*) for bovids; (*Figueirido, Palmqvist & Pérez-Claros, 2009*) for bears; *Gómez Cano, Hernández Fernández & Álvarez Sierra, 2013* for rodents, *Claude et al., 2004* for testudinoids, or *Foth, Rabi & Joyce, 2017* for turtles. This study shows that the three-dimensional shape of the shell of extant turtles, as herein captured using landmarks and semilandmarks curves, allows discriminating with high confidence two primary ecological categories, in particular a terrestrial category, a polyphyletic assemblage that consists of most testudinids and some of the emydids and geoemydids included in our sample, and a highly aquatic category, another polyphyletic assemblage that includes all chelonioids, most trionychids, and some chelydroids included in our sample. All remaining turtles are left behind in a poorly diagnosed, intermediate category which unites an eclectic group of fully terrestrial to highly aquatic turtles with what amounts to a non-specialized continental shell shape. We therefore have confidence in using this method to assess the ecology of fossil turtles with the caveat, however, that it is only possible to recognize two specialized morphotypes.

### Results for fossil turtles

We find the results of our pFDA analyses to be dubious, as the Triassic fossil turtles are not grouping anywhere close to any extant turtle, in contrast to the PCA and LDA, where these turtles plot within the morphospace defined by extant members of the group. This placement of the Triassic turtles as outliers in the pFDA graph could be a direct result of time calibration combined with the phylogenetic placement of these turtles at the base of the turtle tree. This hypothesis was explored with a series of tests, including, among others, use of an ultrametric tree (i.e., all fossils were coded as living in the Present) and use of an artificial outgroup (i.e., an all 0 outgroup, an all 1 outgroup, and an outgroup with random values) with changing ecology (i.e., terrestrial versus unknown). In the plots resulting from the use of an ultrametric tree, the Triassic fossils pool with extant turtles, even though their phylogenetic distance has actually increased (see Fig. S4 and Fig. S5). This makes us question the application of this method on this dataset. The problematic placement of the Triassic fossils as outliers is not solved in any of the six variant analyses using an artificial outgroup, as their position remains mostly unchanged (see Fig. S6 and Fig. S7). As all pFDA performed resulted in an optimal $\lambda = 0$, none of the subsets of

**Table 9** **Prediction of the ecology fossil turtles based on the LDA for SET1.** Results are presented for the analyses using five versus three ecological categories.

|  | Species | Predictions | CAT. | prob. |
|---|---|---|---|---|
| 5 categories | *Plesiochelys bigleri* | fully webbed | 3-4 | 0.95 |
|  | *Proterochersis robusta* | fully webbed | 3-4 | 0.98 |
|  | *Proganochelys quenstedtii* | fully webbed | 3-4 | 0.99 |
| 3 categories | *Plesiochelys bigleri* | intermediate | 1-2 | 0.87 |
|  | *Proterochersis robusta* | intermediate | 1-2 | 0.99 |
|  | *Proganochelys quenstedtii* | intermediate | 1-2 | 0.98 |

the data contain a phylogenetic signal under Brownian motion (*Pagel, 1999*; *Motani & Schmitz, 2011*). This may have led to the curious placements of *Proterochersis robusta* and *Proganochelys quenstedtii*. As shell shape seems to be independent from turtle phylogeny, a phylogenetic correction of the data is unjustified. Consequently, we restrict ourselves to discussing the LDA results only.

## Paleoecology of *Plesiochelys bigleri*

*Plesiochelys bigleri* was included in the study to test the impact of fossils on the study, but also because the paleoecology of plesiochelyids remains poorly resolved as either riverine (*Rütimeyer, 1873*), near-shore marine (*Billon-Bruyat et al., 2005*), or marine (*Bräm, 1965*). This uncertainty is based, in part, on the realization that the sediments that preserve plesiochelyids contain a mixture of continental to marine faunas (*Comment, 2015*), the fact that no complete limbs are yet preserved (*Anquetin, Püntener & Joyce, 2017*), and that the geochemical study of *Billon-Bruyat et al. (2005)* lacks catalog numbers for the specimens included in the study that would allow a verification of their results (*Anquetin, Püntener & Joyce, 2017*).

In the LDA using five categories, *Plesiochelys bigleri* is predicted to be "fully webbed" and plots at the margin of "fully webbed" turtles close to turtles with "extensive webbing". The equivalent analysis using three categories predicts this fossil to be "intermediate," but it plots again within this group towards the margin with "specialized aquatic turtles." These predictions translate into a non-specialized aquatic morphology that is broadly consistent with riverine to costal habitats. Although this does not clarify the ecology of this turtle beyond the debate outlined above, it at least provides independent support for a highly aquatic lifestyle and make the prediction that this animal will reveal to have relatively elongate limbs, but not fully formed flippers.

## Paleoecology of *Proterochersis robusta*

*Proterochersis robusta* has traditionally been thought to have had been a terrestrial turtle (*Fraas, 1913*; *Młynarski, 1976*; *De Lapparent de Broin, 2001*), but this was likely based on the highly domed habitus of the shell combined with the continental sediments from which it was recovered. The study of *Scheyer & Sander (2007)* confirmed this assertion more recently using bone histology, but *Benson et al. (2011)* soon after concluded upon a semi-aquatic lifestyle based on the cross-section of this animal. *Lichtig & Lucas (2017)*

finally concluded upon terrestrial habitat preferences, once again, mostly based on shell ratios that pertain to the doming.

The LDA that utilizes five categories predicts that *Proterochersis robusta* is "fully webbed". It also plots at the margin of "fully webbed", but close to turtles that are "poorly webbed" such as the emydid *Emys blandingii* and the chelid *Platemys platycephala*, which are poor swimmers, but also the geoemydids *Cuora amboinensis* and *Melanochelys trijuga*, which are described as semi-aquatic turtles (*Ernst & Barbour, 1989*). The analysis that utilizes three categories, by contrast, predicts an "intermediate" ecology, which corresponds to a non-specialized shell shape consistent with continental habitat preferences, including fully aquatic, semi-terrestrial, or fully terrestrial. It is interesting to note that this highly domed species does not group with today's highly domed specialized terrestrial tortoises, but rather with the emydid *Emys orbicularis*, and the geoemydids *Mauremys reevesii* and *Heosemys spinosa*, which are also described as semi-aquatic (*Ernst & Barbour, 1989*). We therefore interpret these results as deeply ambiguous but note that depositional environments strongly favor a dry continental setting for this turtle, which is consistent with shell histology, and not contradicted by shell shape either.

### Paleoecology of *Proganochelys quenstedtii*

*Proganochelys quenstedtii* was initially believed to be terrestrial, despite its relatively low domed shell, which was interpreted as being crushed (*Fraas, 1899*; *Jaekel, 1914*). *Gaffney (1990)* noted similarities in the shape of the femur with *Macrochelys temminckii* and concluded upon a possible bottom walking adaptation by reference to the work of *Zug (1971)*. *Joyce & Gauthier (2004)* suggested terrestrial habitat preferences for this taxon based on forelimb proportions, which was soon after confirmed by *Scheyer & Sander (2007)* using bone histology. *Joyce (2015)*, more recently, presented several additional lines of evidence for a terrestrial habitat preference of this taxon, including the presence of osteoderms on the neck and the tail and depositional context, in particular the observation that this turtle is found with continental upland faunas, not intermixed with the rich aquatic low land faunas of the time. *Lichtig & Lucas (2017)*, by contrast, concluded upon semi-aquatic habitat preferences using shell metrics.

The LDA using with five ecological categories predicts for *Proganochelys quenstedtii* to have been "fully webbed" (category 2). The analysis with three ecological categories on the other hand suggests "intermediate" habitat preference, though the species plots together with *Proterochersis robusta* towards the edge of the plot, but once again close to semi-aquatic turtles, such as the testudinoids *Glyptemys insculpta*, *Heosemys spinosa*, and *Emys orbicularis*. In our opinion, the analysis suggests that this turtle has a non-specialized shell shape broadly consistent with continental habitat preferences ranging from fully aquatic to fully terrestrial. The majority of independent sources of information nevertheless still point towards a more dry continental signal.

### Do 2D components perform better than 3D data?

The relative performance of 2D versus 3D data in geometric morphometrics has recently been discussed (*Álvarez & Perez, 2013*; *Cardini, 2014*; *Buser, Sidlauskas & Summers, 2018*;

*Courtenay et al., 2018*; *Otárola-Castillo et al., 2018*; *Hedrick et al., 2019*), but the results are divergent depending on the clade and/or the anatomical body region being studied. This analysis utilized several subsets (SET2 to 4) of the same primary dataset of shell morphology (SET1) of extant and fossil turtles. The confusion matrices and the plots confirm higher accuracy in predicting the known ecology of extant turtles for SET1 and SET2. As such, SET2, which uses the outline of the carapace only, appears to be a better proxy for distinguishing ecological categories in extant turtles than the transverse cross-section (SET3), which were used by *Domokos & Várkonyi (2008)* and *Benson et al. (2011)*. Indeed, the latter was found in this study to show the worst correlation with forelimb webbing and the associated habitat preference. Nevertheless, the complete shell shape (SET1) performs slightly better than the outline shape alone (SET2), suggesting that the full shell is needed to characterize the ecology of a turtle.

## Limits to the study

This study focused on obtaining the 3D shape of a broad set of extant turtles that samples all major clades, but did not consider ontogenetic changes, sexual dimorphism, and variation within genera (see *Rivera, 2008*, for variation within a species). These concerns may be relevant, considering that some extant turtles display much variation during ontogeny and between the sexes (e.g., *Berry & Shine, 1980*; *Pritchard, 2008*; *Vega & Stayton, 2011*). A bigger concern perhaps is that the study only includes few fossil taxa. This was done in part to avoid circularity, but also because intact fossil shells are extremely rare in collections. This has the unfortunate result, however, that shell morphologies not realized by extant turtles for a particular habitat preference or shell morphologies not realized by extant turtles at all are not included in the study, even though they plausibly may have a significant impact. For instance, numerous fossil turtles exist that are believed to have been terrestrial using external data, but that have shell shapes very different from their extant relatives, such as the large, but flat, and often spiked shells of nanhsiungchelyids (e.g., *Hirayama et al., 2001*) or the elongate, but flat shells of sichuanchelyids (*Joyce et al., 2016*). Similarly, numerous taxa thought to be marine, at least by reference to the depositional environment in which they are found, have shells that are similar to freshwater aquatic turtles, such as *Chedighaii barberi* or *Taphrosphys sulcatus* (*Gaffney, Tong & Meylan, 2006*), or display hyperspecialized marine morphologies, such as seen in the thalassochelydians *Achelonia formosa* and *Tropidemys seebachii* (*Joyce & Mäuser, 2020*) or advanced protostegids such as *Archelon ischyros* or *Calcarichelys gemma* (*Wieland, 1903*; *Hooks, 1998*). Inclusion of these fossils, if they ever become available in 3D, would likely render the specialized terrestrial versus specialized marine fields categories used in this study even less diagnostic. The impact of fossils was previously illustrated for turtle skulls by *Foth, Rabi & Joyce (2017)*. Unfortunately, the vast majority of fossils, especially shells, show much taphonomic crushing. In this study, we partially accounted for this by selecting material we felt to be preserved correctly in three dimensions, but we cannot discount subtle plastic deformation. Indeed, a possible additional source of error to our study is usage of a model of *Proganochelys quenstedti*, which was produced as faithfully as possible by reference to the available, crushed fossil

material, but may include subconscious biases of the artist, in addition to taphonomic crushing.

As an alternative to the discriminant analysis we used herein, future studies may wish to focus on explicitly identifying morphologies associated with particular habitat preferences. For instance, we note informally that the tear-drop shape of extant marine turtles and carettochelyids is uniquely associated with highly aquatic animals, that round, but tectate shells seems to be associated with riverine environments, and self-righting shell shapes, as previously proposed (*Domokos & Várkonyi, 2008*) with terrestrial habitats, but that generalized shell shapes can occur everywhere. The identification of specialization may therefore provide better results, than the characterization of the morphospace held by all individuals of a certain ecological category. No matter what, as no single source of ecological information appears to be sufficient for the moment to infer the paleoecology of fossil turtles, we recommend a multi-pronged approach, which includes limb morphology (e.g., *Joyce & Gauthier, 2004*), bone histology (e.g., *Scheyer & Sander, 2007*), isotopic analysis (e.g., *Billon-Bruyat et al., 2005*), depositional environments, cranial morphology (e.g., *Foth, Rabi & Joyce, 2017*), and, if at all, the full morphology of the shell, not just isolated measurements.

## CONCLUSIONS

This study explicitly sought correlations between turtle shell shape and turtle ecology but ended up questioning the utility of shell shape as a proxy for the paleoecology of fossil turtles. Linear discriminant analysis identified two specialized shell shapes that are associated with extant turtles with highly aquatic versus highly terrestrial habitat preferences. Although these correlations could be applied to the fossil record, they are not particularly useful, as the paleoecology of fossil turtles with these shapes is rarely controversial in the first place. Instead, linear discriminant analysis also highlights that the vast majority of extant turtles exhibit an intermediate morphology, regardless of their habitat preferences. Although we did not include fossil turtles to avoid circularity, we presume that their inclusion would further blur the lines, as numerous fossils we perceive to possess this intermediate shell morphotype are otherwise thought to be highly marine and highly terrestrial. From an evolutionary standpoint, this indicates that the shape of the turtle shell is likely controlled by factors unrelated to ecology. We urge caution for assessing the paleoecology of fossil turtles using shell shape alone. Most importantly, the commonly propagated rule of thumb that a domed shell corresponds to terrestrial ecology, while a flattened one suggests an aquatic lifestyle, should be avoided, as many turtles perceived to be highly domed have an aquatic ecology.

**Institutional abbreviations**

| | |
|---|---|
| **FMNH** | Field Museum of Natural History Chicago, Illinois, USA |
| **NMB** | Naturhistorisches Museum Basel, Switzerland |
| **MHNF** | Museum d'Histoire Naturelle de Fribourg, Switzerland |
| **SMNS** | Staatliches Museum für Naturkunde Stuttgart, Germany |
| **MJSN** | JURASSICA Museum, formerly Musée Jurassien des Sciences Naturelles, Porrentruy, Switzerland |

## ACKNOWLEDGEMENTS

We thank numerous people and institutions for providing us with access to specimens in their care, in particular Alan Resetar and Joshua Mata (FMNH), Loic Costeur, Eduard Stöckli, and Ambros Hänggi (NMB), Emanuel Gerber (MHNF), Erin Maxwell and Rainer Schoch (SMNS). We are indebted to Jérémy Anquetin and Irena Raselli (Jurassica Museum) for providing us with a 3D model of *Plesiochelys bigleri* and to Mónica Angulo-Bedoya (EAFIT) for providing us with the 3D model of *Carettochelys insculpta*. We thank Olivia Plateau and Bastien Mennecart for useful discussions and comments. We are grateful to Doug Boyer for his help creating the MorphoSource project that holds the 3D models used in this study. We thank Christine Böhmer, Julien Claude, and an anonymous reviewer for comments that greatly improved the manuscript.

### Funding

This project was funded by the Department of Geosciences of the University of Fribourg and by a grant from the Swiss National Science Foundation to Walter G. Joyce (SNF 200021_178780/1). There was no additional external funding received for this study. The funders had no role in study design, data collection and analysis, decision to publish, or preparation of the manuscript.

### Grant Disclosures

The following grant information was disclosed by the authors:
Department of Geosciences of the University of Fribourg.
Swiss National Science Foundation.

### Competing Interests

The authors declare there are no competing interests.

### Author Contributions

- Dziomber Laura conceived and designed the experiments, reconstructed 3D models, performed the experiments, analyzed the data, prepared figures and/or tables, authored or reviewed drafts of the paper, and approved the final draft.
- Joyce G. Walter  and Foth Christian  conceived and designed the experiments, authored or reviewed drafts of the paper, and approved the final draft.

### Data Availability

    Raw data, intermediate results and code are available in the Supplemental Files.
    The 3D models are available at Morphosource project 1028: https://www.morphosource.org/Detail/ProjectDetail/Show/project_id/1038
      - Chelus fimbriatus, M64746-116408 https://doi.org/10.17602/M2/M116408
      - Platemys platycephala, M64747-116409 https://doi.org/10.17602/M2/M116409

- Mesoclemmys dahli, M64749-116411 https://doi.org/10.17602/M2/M116411
- Phrynops tuberosus, M64750-116412 https://doi.org/10.17602/M2/M116412
- Elseya novaeguineae, M64751-116413 https://doi.org/10.17602/M2/M116413
- Emydura macquarii, M64752-116414 https://doi.org/10.17602/M2/M116414
- Hydromedusa tectifera, M64753-116415 https://doi.org/10.17602/M2/M116415
- Chelodina oblonga, M64754-116416 https://doi.org/10.17602/M2/M116416
- Chelydra serpentina, M64755-116419 https://doi.org/10.17602/M2/M116419
- Dermatemys mawii, M64756-116420 https://doi.org/10.17602/M2/M116420
- Dermochelys coriacea, M64757-116421 https://doi.org/10.17602/M2/M116421
- Trachemys scripta, M64758-116422 https://doi.org/10.17602/M2/M116422
- Terrapene carolina, M64759-116423 https://doi.org/10.17602/M2/M116423
- Clemmys guttata M64760-116424 https://doi.org/10.17602/M2/M116424
- Emys orbicularis, M64761-116425 https://doi.org/10.17602/M2/M116425
- Glyptemys insculpta, M64762-116426 https://doi.org/10.17602/M2/M116426
- Emydoidea blandingii, M64763-116427 https://doi.org/10.17602/M2/M116427
- Deirochelys reticularia, M64764-116428  https://doi.org/10.17602/M2/M116428
- Graptemys geographica, M64765-116429 https://doi.org/10.17602/M2/M116429
- Malaclemys terrapin, M64766-116430 https://doi.org/10.17602/M2/M116430
- Chrysemys picta M64767-116431 https://doi.org/10.17602/M2/M116431
- Actinemys marmorata, M64768-116432 https://doi.org/10.17602/M2/M116432
- Geoemyda spengleri, M64770-116434 https://doi.org/10.17602/M2/M116434
- Vijayachelys silvatica, M64771-116435 https://doi.org/10.17602/M2/M116435
- Rhinoclemmys annulata, M64772-116436 https://doi.org/10.17602/M2/M116436
- Cuora amboinensis, M64773-116437 https://doi.org/10.17602/M2/M116437
- Cyclemys dentata, M64774-116438 https://doi.org/10.17602/M2/M116438
- Heosemys spinosa, M64775-116439 https://doi.org/10.17602/M2/M116439
- Mauremys reevesii, M64776-116440 https://doi.org/10.17602/M2/M116440
- Melanochelys trijuga, M64777-116441 https://doi.org/10.17602/M2/M116441
- Notochelys platynota, M64782-116446 https://doi.org/10.17602/M2/M116446
- Orlitia borneensis, M64783-116447 https://doi.org/10.17602/M2/M116447
- Pangshura tentoria, M64784-116448 https://doi.org/10.17602/M2/M116448
- Sacalia quadriocellata, M64785-116449 https://doi.org/10.17602/M2/M116449
- Malayemys subtrijuga, M64786-116450 https://doi.org/10.17602/M2/M116450
- Morenia petersi, M64787-116451 https://doi.org/10.17602/M2/M116451
- Batagur dhongoka, M64788-116452 https://doi.org/10.17602/M2/M116452
- Claudius angustatus, M64789-116453 https://doi.org/10.17602/M2/M116453
- Staurotypus triporcatus, M64790-116454 https://doi.org/10.17602/M2/M116454
- Sternotherus odoratus, M64791-116455 https://doi.org/10.17602/M2/M116455
- Kinosternon baurii, M64792-116456 https://doi.org/10.17602/M2/M116456
- Pelusios sinuatus, M64793-116457 https://doi.org/10.17602/M2/M116457
- Pelomedusa subrufa, M64794-116458 https://doi.org/10.17602/M2/M116458
- Platysternon megacephalum, M64795-116459  https://doi.org/10.17602/M2/M116459
- Podocnemis vogli, M64796-116460 https://doi.org/10.17602/M2/M116460

- Astrochelys radiata, M64798-116462 https://doi.org/10.17602/M2/M116462
- Chelonoidis carbonaria, M64800-116464 https://doi.org/10.17602/M2/M116464
- Chersina angulata, M64801-116465 https://doi.org/10.17602/M2/M116465
- Geochelone elegans, M64802-116466 https://doi.org/10.17602/M2/M116466
- Gopherus polyphemus, M64805-116469 https://doi.org/10.17602/M2/M116469
- Homopus femoralis, M64806-116470  https://doi.org/10.17602/M2/M116470
- Indotestudo elongata, M64808-116472 https://doi.org/10.17602/M2/M116472
- Kinixys belliana, M64810-116474 https://doi.org/10.17602/M2/M116474
- Malacochersus tornieri, M64811-116475 https://doi.org/10.17602/M2/M116475
- Manouria impressa, M64812-116476 https://doi.org/10.17602/M2/M116476
- Psammobates tentorius, M64815-116479 https://doi.org/10.17602/M2/M116479
- Pyxis arachnoides, M64817-116481 https://doi.org/10.17602/M2/M116481
- Stigmochelys pardalis, M64819-116483 https://doi.org/10.17602/M2/M116483
- Testudo graeca, M64820-116484 https://doi.org/10.17602/M2/M116484
- Dogania subplana, M64824-116488 https://doi.org/10.17602/M2/M116488
- Pelodiscus sinensis, M64826-116490 https://doi.org/10.17602/M2/M116490
- Rafetus euphraticus, M64827-116491 https://doi.org/10.17602/M2/M116491
- Apalone mutica, M64828-116492 https://doi.org/10.17602/M2/M116492
- Lissemys punctata, M64829-116493 https://doi.org/10.17602/M2/M116493
- Proganochelys quenstedtii, M64830-116494 https://doi.org/10.17602/M2/M116494
- Proterochersis robusta, M64831-116495 https://doi.org/10.17602/M2/M116495
- Chelonia mydas, M64778-116567 https://doi.org/10.17602/M2/M116567
- Macrochelys temminckii, M64779-116443 https://doi.org/10.17602/M2/M116443
- Eretmochelys imbricata, M64780-116444 https://doi.org/10.17602/M2/M116444
- Caretta caretta, M64781-116445 https://doi.org/10.17602/M2/M116445.

## Supplemental Information

Supplemental information for this article can be found online at http://dx.doi.org/10.7717/peerj.10490#supplemental-information.

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
