# Peer review of "The ecomorphology of the shell of extant turtles and its applications for fossil turtles"

_PeerJ, doi:10.7717/peerj.10490_

## Round 0.1 · original submission · Major Revisions

I think that this is a strong paper with a lot of potential. Please see the comments presented by the reviewers and be especially careful of wording in your methods section (see reviewer 3 comments).

Both reviewers mention that the supplemental information has metadata, but not primary data. So long as the scans you used for your analyses are not going to be used in a paper that is forthcoming, I would strongly encourage you to publish them online. If the scans will be used, it would still be good to publish your landmark files.

Please let me know if you have any questions and thank you for your submission.

Reviewer 1 ·

Basic reporting

This study is based on extensive statistical and morphometric analyses. Since I am not an expert in the methods employed, I find myself unable to competently review this submission.

Experimental design

NA

Validity of the findings

NA

Additional comments

NA

·

Basic reporting

This study investigated the relationship between shell morphology and ecology in turtles in order to evaluate if the turtle shell can be used to reliably infer the paleoecology of extinct taxa. A 3D geometric morphometric approach was used to quantify the shape of the shells in a broad sample of extant species (N=69) that represent all major turtle clades and three fossil turtles. Based on their results, the authors conclude that there is “much overlap between habitat groups”, but they detected “significant differences” between the extremes (highly specialized terrestrial and aquatic turtles). Indeed, the secondary analysis using fewer ecological categories revealed similar results with a lower misclassification rate.

Basic reporting
In my opinion, the introduction is globally sound. The material and methods section provides all necessary information on collected data and used analytical methods (but refer to specific comments below). The results are presented in a clear manner, making it easy to follow the descriptions. The discussion section could benefit by including some additional, relevant references (refer to specific comments below).

The figures are straightforward. Nevertheless, I would like to mention that some colleagues prefer to use color-blind-friendly colors (or, alternatively, to use different symbols in addition to color) to make their graphs accessible to everyone. The tables are informative. The data sampling appears reasonable. The supplementary material is ok, but I recommend to think about providing the raw data of the (semi-)landmark coordinates placed on each specimen as well as the R code. For instance, refer to other works published in PeerJ:
- Ascarrunz et al. 2019 PeerJ Supplementary material “raw landmark data in tps format”, doi: 10.7717/peerj.7476/supp-7 and “R scripts”, doi: 10.7717/peerj.7476/supp-8
- Mallet et al. 2019 PeerJ Supplementary material “raw data tps file”, doi: 10.7717/peerj.7647/supp-9 and “R code used for the analysis”, doi: 10.7717/peerj.7647/supp-10

Experimental design

The present study is within the scope of the journal. The methods are sound. The authors explain that they investigate the relation between shell shape and ecology, but I miss some specific hypotheses; in particular, because there are previous studies that examined this relation as well (as correctly mentioned by the authors).

Validity of the findings

The results appear valid and meaningful. We lack the raw data to be able to replicate the analyses, but the intermediate data (results from PCA etc.) are provided in the supplementary material. The paper discusses the limits of the present study. The work could benefit by including some important references (refer to specific comments below).

Additional comments

Specific comments (alternatively refer to attached document)

Overall, this is a very interesting and comprehensive work! I enjoyed reading the manuscript. I do have some suggestions and comments that should be addressed before acceptance.

Abstract

[30 ff] “Principal component analysis (PCA) highlights much overlap between habitat groups. Phylogenetic flexible discriminant analysis (pFDA), on the other hand, suggests differences, but only between highly specialized terrestrial turtles, highly specialized aquatic turtles, […].”
In my opinion, the differences between the extremes are also evident in the PCA. Thus, I would not state “on the other hand”. I would recommend to just leave the “on the other hand” out.

Introduction

[56] “The resulting bones of the carapace of a typical turtle are called […].”
Not sure if it is very informative to name the individual bones (at least in its current form) without giving any further information, such as a figure or some other detail, to explain why this is important for the present study.

[78] “Given that correlations appear to exist between shell shape and ecology […].”
Here or maybe at the end of the introduction, I would have expected some precise hypotheses (based on functional assumptions for example). What exactly do you expect from your study?

[123 ff] “The correlations observed among extant turtles are then applied to the Triassic turtles […].”
It’s a bit misleading that the Late Jurassic Plesiochelys bigleri is not mentioned here.

Material and Methods

[190] […] inguinal buttress (landmarks 6 and 7) […] (Fig. 1)
I am wondering if landmarks 6 and 7 (visible in Fig. 1E) should be shown in Fig. 1C as well? Furthermore, I have the impression that landmark 1 is not positioned exactly in Fig. 1A. Do you mind checking this?
[193] […] twelve semi-landmark curves […] (curves C1 and C2) […]
I suggest to name the curves in figure 1 (as it is done in figure 2). It would then be easier to follow the description in the text.

[193] […] twelve semi-landmark curves […] (curves C1 and C2) […]; Fig. 3, Tab. 1
The reader can imagine that the ecological category numbers (“Ecol 0-4” in table 1 and “Cat0-4” in Table 3) correspond to the webbing types (A-E) displayed in figure 3, but it would be useful to indicate the numbers in figure 3 as well to avoid any misunderstandings. Furthermore, I would avoid writing “Cat” in table 3 because you do not use “Cat” in table 1. It’s unnecessary and may confuse some readers.

[235 ff] […] The PCA was computed using the function PlotTangentSpace in the software package Geomorph […] and the R software and language […]”
Please consider revising the sentence. Geomorph is a package in R.

Results
[275] […] For SET1 (Fig. 6A) […]
Fossils are indicated as diamonds in the legend of figure 6, but shown as stars in some PCA graphs. Please correct that.

[278] […] (negative PCs) […]
Since you’re describing PC1 here, I suggest to write “negative PC1 scores” instead in order to be precise. Dito for the other PCA descriptions (SET1-4).

[279] […] categorized by the presence of flippers (4) are scattered […]
I suppose you mean “category 4”. I strongly recommend to make this homogenous throughout the whole (!) manuscript (including the tables and images). Refer also to my previous comment on line 193, Fig. 3 and Table 1.

[331] This is perhaps a result of the tear-drop-shaped shells of these taxa.
In my opinion, this sentence should not be part of the results section, but of the discussion section.

Discussion

[408 ff] “Discussion”
The authors should have a look at the following references. In my opinion, they are important to be considered in the present work:

Rivera (2008) Ecomorphological variation in shell shape of the freshwater turtle Pseudemys concinna inhabiting different aquatic flow regimes. Integrative and Comparative Biology. doi:10.1093/icb/icn088
>> (…) the carapace and plastron show significant morphological differences between habitats characterized by slow-flowing (i.e., lentic) and fast-flowing (i.e., lotic) water. (…) Of the two shell components (carapace and plastron), the carapace shows greater divergence between habitats (…)
Stayton (2011) Biomechanics on the half shell: functional performance influences patterns of morphological variation in the emydid turtle carapace. Zoology. doi: 10.1016/j.zool.2011.03.002

>> Aquatic turtle shells differ in shape from terrestrial turtle shells and are characterized by lower frontal areas and presumably lower drag. Terrestrial turtle shells are stronger than those of aquatic turtles; many-to-one mapping of morphology to function does not entirely mitigate a functional trade-off between mechanical strength and hydrodynamic performance. (…) Given that the shells of terrestrial turtles perform fewer functions than those of aquatic turtles (terrestrial shells are not under selective pressure to improve hydrodynamic performance) (…)

[422] […] possible to recognized two […]
Typo: “recognize” (without the “d”)

[492 f] […] of the same primary dataset of shell (SET1) of extant and fossil turtles.
Do you mean “dataset of shell morphology” or maybe “shells”?

·

Basic reporting

This is a new paper using 3D data to estimate the relationship between turtle shell shape and ecology. It is applied to fossils. The dataset is new and the shape variation analyzed is original. I was overall interested and curious by the results obtained. Gross methodology is fine but analyses can be refined in details. In view of the obtained results the discussion and result section sound.
The paper can be slightly restructured, methods should be clarified or corrected, and depending on results and discussion reworked.

Experimental design

Methods should be reworked see general comments.

Validity of the findings

See general comment (make sure to use jacknife for estimating whether your categories are well discriminated)

Additional comments

This paper is an original study but I suggest several modifications to improve it.


1. Problems regarding R and geomorph, phrasing 'morphometrics'
- geomorph package is badly spelled: geomorph not Geomorph

- I hope that you are not using an R version as old as 2013 since several things were improved since and especially because geomorph packages has seen important changes including sliding semi landmark procedures, please update your reference.

- proCD.lm does not exist in geomorph (it is badly spelled, and it is deprecated in current versions); please update.

l. 236: The PCA was computed using the function PlotTangentSpace in the software package Geomorph (Adams & Otarola-Castillo, 2013) and the R software and language (R Core Team, 2013). -> rephrase the end :
PlotTangentSpace of the geomorph package (Adams & Otarola-Castillo, 2013) in R (R Core Team, 20??)
-> rephrase also the begining: PlotTangentSpace do not only present pca of superimposed coordinates but also project observation onto the tangent shape space by performing an orthogonal projection (look at the package online help).


2. Potential issues regarding morphometric procedures and statistical analyses

2.1.Allometries :
I suggest testing allometry by directly applying the multivariate test rather than by looking at relationship between prediction and actual data. It is much easier. geomorph can do it or you can also simply do it by looking at Claude, 2008 (Morphometrics volume) or 2013 (Hystrix volume). It is actually what you do.
Indeed l.271: anova permutation -> should be in methods: it is actually how you tested the multvariate regression (see remark above).
In doing so, you can also represent the allometric component (shape variation between large and small individuals). I, however, guess that it is likely biased: the largest species is Dermochelys but for that species you have been using a very small and juvenile individual.
l.272-273: Note that correcting for allometries is not obligatory, it depends on the scope of the study: some studies might be interested in shape differences, including those due to allometries !!!.

2.2.Superimposition procedure:
As you are sliding 3D semi landmarks, it is important to say whether you slided them in minimizing the procrustes distance or the bending energy. Both methods can yield rather different results, especially when there is a lot of sliders (it is the case here).

2.3. Ecological categorization:
Ecological categories are rather important in numbers by comparison to previous studies, note that, however that the classification is sometimes complicated and still uncertain: For emydines, the cut between 1 and 2 is sometime difficult: see for instance Emys that I could have classified as more webbed than Glyptemys.

l.380: I suggest modifying the title of that section and reorganizing your text. It is confusing as it refer to methods and results. I suggest to reorganize the text in agreement. You should also re run the LDA analyses with these new grouping. Say in your methods that you tried two different categorisations, explain them here, and then present the results for LDA and pFDA.

2.4: ecological categories and discrimnant analysies
As far as I know, discriminant analysis do not recognize the gradient between all the categories but consider them as independent. In that case, increasing categories could be related to a loss of discrimination between groups. This may explain why discrimination is not so strong in the present analysis.
you simplified your categories in 3 rather than 5 but did you find a similar result in binarizing your factor ?

2.5. Important errors regarding PCA
l.234 : "variables that contain a true biological meaning by using orthogonal transformation (Karamizadeh et al., 2013)" -> please remove that, PC axes are no biological axes: they are purely mathematical axes of maxima of variance covariance (they strictly depend on statistical sampling not on biology). PCA is just a rotation of the data; it is a way to represent multivariate ordination (no more), look at older publication if you have doubts: Reyment, Joliffe....Etc.
l. 275: Principle -> OHHHHH NO!

2.6. LDA and PCA
l. 239: which distinguishes morphological differences between groups (McLachlan, 2004). LDA uses a similar approach compared to PCA but identifies the axes that maximize the separation between multiple classes
-> update reference regarding lda (you can use a reference closer to the original, or some more general textbook, lda is a technique much older than the 21 century).
LDA is not similar to PCA. It looks at axes that maximize intergroup variation and minimize intragroup variation. It is therefore not working on the Variance covariance matrix of the data but on something else (for fast explanation, regarding morphometrics you can return to Claude 2013 or 2008).

2.7. The Brocken Stick method is not from Jackson 1993. Please cite the original citation.
There is no obligation to select only the first PCs axes for the LDA. But you have indeed to remove shape variables because the number of variables in your data set is much larger than the number of observation and categories. Keep in mind that the more you reduce your data set, the more you lose information. However the more variables you keep, the more your result will be sample dependent.
Note that you have several categories but your Brocken still model sometime reduce the shape space to about the same number of dimensions the number of categories, it is therefore not surprising to have a discrimination that is soso, what if you include axes up to 90% or 95 % of total variance ?

2.8. Confusion table and predictive power:
Confusion tables are interesting but more easy to interpret after a one leave out cross validation for avoiding to be sample dependent. You have to estimate it with the argument CV=TRUE when you use lda function. It is not said whether you have been using this in the text.

Discussion part about 2D and 3D data, ok only if your confusion table are based using jacknifing methods (one leave out cross validation). If not, it is just normal to see a better score of 3D because there is an inflation of data by comparison to observation in that case.

3. Data availability and publishing 3D models
It would be good to publish the 3D models and eventually landmark data sets as this represented some important work, you can think about Morphomuseum journal for it (it is free access, and free of charge for authors).

4. How to be sure about the problem of fossil deformation ? How did you manage for that ?

5. discriminant analysis, pca and related shape variation

Note that FDA cannot be easily interpreted in terms of morphology: it does not work on a linear combination of original variables. It is therefore hard to identify features that discriminate groups and to depict discrimant axes in terms of shape differences (landmark data are indeed all inter-dependent). By contrast LDA allows you to draw shape differences along axes. Claude 2008, 2013 explain how to do it in R and released several examples with scripts. It seems that you do this at least for your PCAs, but it is not explained how you computed extreme shape on axes (maybe based on these publications ??). I suggest to do that also for LDA.

6. pFDA, comparative method and position of fossils in the tree:
It is certainly not a good idea to use an ultrametric tree especially if you are using a method that is based on GLS regression such as pFDA, your fossil should be set in time in the tree. This is especially important because some of your fossil are close to the root of the tree. you can not consider that Proganochelys evolved independtly from the root for more than 200 My. This is not realistic. This should be corrected.

7. Interpretation of variation along PCs
l. 277: size of the plastron -> relative size of the plastron (GPA scale everything to unit centroid size, you can not have size differences at this step). Note that this relationship between carapace doming and plastron size is an bias of the method (it scales everything to unit centroid size).
l.283: avoid speaking about clustering when you are just looking at the first PCA plan (less than 42% of total variance): we have absolutely no idea of what is happening on other axes. If you want to know how data are clustered you must use clustering methods.

8. Identifying shape parameters that discriminate among groups:
For the LDA at least you can represent shape variation along axes as for the PCA, see Claude 2008, 2013 again for scripts in R. This can help you to further discuss about the morphological features that discriminate among groups. (See however my remark about binarization because LDA or FDA do not see the categories 1 -> 5 as a continuum). This is particularly important because in the last chapter of the study you speak about things that allows to differentiate groups, but nowhere you characterize these morphological differences.


9. Other
l.276: here and elsewhere in the text at several places: check the punctuation: "explains, 28.5%" -> explains 28.5%

In the discussion, you should also speak about the prediction of the LDA and not restrict it to the pFDA. It is maybe not as advanced as pFDA but it is interesting as well.

Julien CLAUDE, Institut des Sciences de l'Evolution de Montpellier, FRANCE

---

## Round 0.2 · Minor Revisions

Thank you for your submission. This is a very nicely done manuscript and I enjoyed reading it. Both reviewers and I agree that this manuscript is nearly ready for publication. In addition to the few changes the reviewers suggest, I have a number of changes (mostly grammar/spelling) that should be incorporated into the next version.

When you submit your revised manuscript, please submit a reviewer response, tracked changes version of your mansucript, and a clean version. Based on reviewer comments, the manuscript will likely not need to go out to review again.

Best,

Brandon
* * *
Editor Comments:

Line 38: ‘overall too weak to uncover using shell..’

Line 54: This is from an anatomical perspective, not an evolutionary one.

Line 98: ‘measurements’

Line 146: Is there a reference for sex only affecting shell shape in a subtle manner?

Line 235: Delete this sentence about GPA. It’s oddly worded and unnecessary.

Line 241: capitalize ‘ANOVA’ and should be ‘analysis’. Also line 278.

Line 242: ‘principal’. Also line 283.

Line 252: ‘assigned’

Line 253: ‘MASS and was used..’

Line 257: I wouldn’t say they’re less biased so much as that they factor phylogenetic signal into the results. Reword this.

Line 269: lambda equals 1 gives a phylogenetic signal following Brownian motion. I am not sure what a ‘pure phylogenetic signal’ is

Line 319: ‘correspond’

Line 346: ‘dependent’

Line 363: probably should be ‘plots’?

Line 448-449: Is this likely a phylogenetic signal? Can you add a few sentences here?

Line 467: Question the method or question the utility of the method with your data?

Line 470: You could also check Blomberg’s K for multidimensional data (see Adams, 2014), which would tell you if your shape data contain a phylogenetic signal. The function physignal is in geomorph. It might be more reliable than the pFDA lambda estimation (although they would theoretically agree)

Line 485: ‘verification’ instead of ‘verifying’

Line 516: What do you mean by ‘shell histology for the moment’? Is there an ongoing study of shell histology? I would suggest deleting ‘for the moment’

Line 557: What do you mean by ‘time constraints’? Specimen access? I guess most of these factors would be included in analysis and would only take the time that the analysis takes. Can you clarify this a bit more?

Line 573: Update this citation if possible.

Line 615: Fix the grammar in your last sentence.

Figure 6 in my version has the 3D outlines overlapping with the values on the axes. Make sure this isn’t the case for publication.

·

Basic reporting

The authors have carefully revised their original manuscript, taking into account all questions and remarks. In my opinion, the revised version of the manuscript is ready for publication after the following minor issues are addressed:

[46] “an armored body plans”
Delete the “s”.

Figure 1
The figure is improved by the labeling of the anatomical parts, but there is no explanation of the abbreviations. Please consider adding this information in the figure caption.

In general, check for typos and try to write terms always in the same manner (e.g., “linear discriminant analysis” vs. “Linear Discriminant Analysis” etc.).

Experimental design

No comment.

Validity of the findings

No comment.

·

Basic reporting

The ms was well improved since the last version. I have just three minor remarks. See general comments.

Experimental design

no comment

Validity of the findings

no comment

Additional comments

l. 235: add position as gpa also remove translation

l. 238: there is no need for transforming size by its log. Allometry is the relationship between shape and size, not between shape and log size. We are not working on linear measurements (where which require the allometric equation with logs) here but on shape data (defined by landmarks).


l. 280: should precise the nature of the allometry -> interspecific allometric signal

---

## Round 0.3 · accepted · Accept

Thank you for your submission! This was a very interesting paper and I enjoyed reading it. I have moved it forward to the production stage and am excited to see it published.